JOURNAL OF
Neuroscience Research

# ACSA-2 and GLAST classify subpopulations of multipotent and glial-restricted cerebellar precursors

Christina Geraldine Kantzer[1,2] (iD) | Elena Parmigiani[3,4,5] (iD) | Valentina Cerrato[3,4,6] | Stefan Tomiuk[1] | Michail Knauel[1] | Melanie Jungblut[1] (iD) | Annalisa Buffo[3,4] | Andreas Bosio[1]

[1]Miltenyi Biotec B.V. & Co. KG, Bergisch Gladbach, Germany

[2]Department of Cell and Molecular Biology, Karolinska Institute, Solna, Sweden

[3]Department of Neuroscience Rita Levi-Montalcini, University of Turin, Turin, Italy

[4]Neuroscience Institute Cavalieri Ottolenghi, Orbassano, Italy

[5]Department of Biomedicine, University of Basel, Basel, Switzerland

[6]Department of Fundamental Neurosciences, University of Lausanne, Lausanne, Switzerland

**Correspondence**
Annalisa Buffo, Department of Neuroscience Rita Levi-Montalcini, University of Turin, I-10126 Turin, Italy.
Email: annalisa.buffo@unito.it

Andreas Bosio, Miltenyi Biotec B.V. & Co. KG, Friedrich-Ebert-Straße 68, 51429 Bergisch Gladbach, Germany.
Email: AndreasBo@miltenyibiotec.de

**Funding information**
Local funds of the University of Turin to AB; Ministero dell'Istruzione, dell'Università e della Ricerca—MIUR project "Dipartimenti di Eccellenza 2018–2022" to Dept. of Neuroscience "Rita Levi Montalcini"; IBRO-PERC in Europe Short Stay fellowship to VC; Umberto Veronesi Foundation to VC

## Abstract

The formation of the cerebellum is highly coordinated to obtain its characteristic morphology and all cerebellar cell types. During mouse postnatal development, cerebellar progenitors with astroglial-like characteristics generate mainly astrocytes and oligodendrocytes. However, a subset of astroglial-like progenitors found in the prospective white matter (PWM) produces astroglia and interneurons. Characterizing these cerebellar astroglia-like progenitors and distinguishing their developmental fates is still elusive. Here, we reveal that astrocyte cell surface antigen-2 (ACSA-2), lately identified as ATPase, Na+/K+ transporting, beta 2 polypeptide, is expressed by glial precursors throughout postnatal cerebellar development. In contrast to common astrocyte markers, ACSA-2 appears on PWM cells but is absent on Bergmann glia (BG) precursors. In the adult cerebellum, ACSA-2 is broadly expressed extending to velate astrocytes in the granular layer, white matter astrocytes, and to a lesser extent to BG. Cell transplantation and transcriptomic analysis revealed that marker staining discriminates two postnatal progenitor pools. One subset is defined by the co-expression of ACSA-2 and GLAST and the expression of markers typical of parenchymal astrocytes. These are PWM precursors that are exclusively gliogenic. They produce predominantly white matter and granular layer astrocytes. Another subset is constituted by GLAST positive/ACSA-2 negative precursors that express neurogenic and BG-like progenitor genes. This population displays multipotency and gives rise to interneurons besides all glial types, including BG. In conclusion, this work reports about ACSA-2, a marker that in combination with GLAST enables for the discrimination and isolation of multipotent and glia-committed progenitors, which generate different types of cerebellar astrocytes.

**KEYWORDS**
ACSA-2, astrocytes, Bergmann glia, cerebellum, prospective white matter, RRID:AB_10000325, RRID:AB_10000343, RRID:AB_10013382, RRID:AB_1036062,

Edited by Stephen Crocker, Junie Warrington, and David McArthur. Reviewed by Cory Willis and Richard Milner.

Christina Geraldine Kantzer and Elena Parmigiani contributed equally to this work.

--------

RRID:AB_10829302, RRID:AB_10829314, RRID:AB_1566510, RRID:AB_2113602,
RRID:AB_221569, RRID:AB_2299035, RRID:AB_2307313, RRID:AB_244365,
RRID:AB_244373, RRID:AB_2651192, RRID:AB_2655070, RRID:AB_2655072,
RRID:AB_2655591, RRID:AB_2660782, RRID:AB_2660783, RRID:AB_2727361,
RRID:AB_2727421, RRID:AB_2727423, RRID:AB_442102, RRID:AB_477499,
RRID:AB_641021, RRID:AB_871621, RRID:AB_871623, RRID:AB_90949, RRID:AB_91789,
RRID:ISMR_CARD:290, RRID:SCR_001905, RRID:SCR_001915, RRID:SCR_002798,
RRID:SCR_003070, RRID:SCR_005012

# 1 | INTRODUCTION

Astrocytes, the most abundant type of glia cells of the central nervous system (Kettenmann & Ransom, 2005), preserve brain function and execute a variety of neuro-protective mechanisms (Ben Haim & Rowitch, 2017; Bylicky et al., 2018). Although their physiological properties are very well studied (Verkhratsky & Nedergaard, 2018), their functional and molecular properties during development, homeostasis, aging, and disease are not fully understood yet (Becerra-Calixto & Cardona-Gomez, 2017; John Lin et al., 2017; Matias et al., 2019; Miller, 2018). A versatile model to study the ontogeny of astrocytes is the cerebellum; hence its four-layered structure presents distinct major astroglial phenotypes. Cerebellar cell types are clearly distinguishable by morphology, molecular features, and layering: the innermost layer, the white matter (WM), and the intermediate layer, the granular cell layer (GL), harbor parenchymal astrocytes including fibrous and velate protoplasmic astrocytes, respectively. The Purkinje cell layer (PCL) marks the border between the GL and the outermost layer, the molecular layer (ML). In the ML unipolar Bergmann glia (BG) and Fañanas cells can be found (Buffo & Rossi, 2013; Goertzen & Veh, 2018).

Astroglial precursors in the nascent cerebellum are poorly understood. Astrogliogenesis occurs in two secondary proliferative niches: the prospective white matter (PWM) and the PCL. Descendants of the PCL are BG and velate granular layer astrocytes (Cerrato et al., 2018). Astroglia-like cells in the PWM are multipotent and develop interneurons and WM astrocytes at a defined period of the postnatal age (Parmigiani et al., 2015).

The astrocyte cell surface antigen-2 (ACSA-2) (Kantzer et al., 2017) binds to an extracellular epitope of the ATPase, Na$^+$/K$^+$ transporting, beta 2 polypeptide (ATP1B2) (Batiuk et al., 2017)—also known as the adhesion molecule on glia (Antonicek & Schachner, 1988; Pagliusi et al., 1990). It allows for specific identification and isolation of living astrocytes from wild-type mice. Although ACSA-2 is predominantly co-expressed with the pan astrocyte marker GLAST (Kantzer et al., 2017) its expression pattern appears unique in the adult cerebellum (Batiuk et al., 2017; Kantzer et al., 2017).

Here we describe the expression of ACSA-2 in defined astroglial-like progenitors throughout the development of the cerebellum. Immunophenotyping, transcriptomic data, and transplantation assays demonstrate that ACSA-2 marks a subpopulation of GLAST$^+$ cells and thus discriminates two distinct populations of multipotent

**Significance**

The developing cerebellum comprises a variety of glial precursor populations. Distinguishing these populations will be beneficial to better understand the role of glial cells during cerebellar development and their impact on developmental diseases. By using astrocyte cell surface antigen-2 (ACSA-2) in combination with GLAST we developed a cell surface marker code that allows for the separation of multipotent versus parenchymal astrocytes-restricted precursors from the neonatal murine cerebellum. Consequently, this makes ACSA-2 a powerful tool to study differences in cerebellar glia not only in homeostasis but also during development and disease.

and glia-committed progenitors, which generate distinct types of cerebellar astrocytes.

# 2 | MATERIALS AND METHODS

## 2.1 | Experimental animals

Experimental procedures were performed on wild-type mice (C57BL/6 background), on β-actin–green fluorescent protein (GFP) (Okabe et al., 1997) (RRID:ISMR_CARD:290) and human glial fibrillary acidic protein (hGFAP)-GFP transgenic mice (Zhuo et al., 1997). Animals were maintained under specific pathogen-free conditions according to the recommendations of the Federation of European Laboratory Animal Science Association.

All procedures were carried out in accordance with the European Communities Council Directive European Communities Council (86/609/EEC) and (2010/63/EU), the NIH guidelines and the Italian Law for Care and Use of Experimental Animals, (DL26/14), and were approved by the Italian Ministry of Health and the Bioethical Committee of the University of Turin.

## 2.2 | Tissue dissociation

Murine neonatal cerebellar tissue, of either sex, was dissociated using the Neural Tissue Dissociation Kit – Postnatal Neurons with

trypsin (Miltenyi Biotec) in combination with the gentleMACS™ Octo Dissociator with heaters. Dissociated tissue was passed through a 100 μm cell strainer (BD) and pelleted by 10 min centrifugation at 300 g at room temperature (RT). The cell pellet was resuspended in D-PBS buffer (Lonza) and processed immediately.

## 2.3 | Flow cytometry analysis

Murine single-cell suspensions, of either sex, were obtained from cerebellar tissue, incubated with FcR Blocking Reagent mouse (Miltenyi Biotec) to prevent unspecific antibody binding and then stained with fluorochrome-conjugated antibodies (compare also Table 1): Anti-ACSA-2-((FITC) Cat# 130-116-243 RRID:AB_2727421), Anti-ACSA-2-((APC) Cat# 130-116-245 RRID:AB_2727423), Anti-ACSA-2-((PE) Cat# 130-116-141 RRID:AB_2727361), Anti-GLAST (ACSA-1)-((APC) Cat# 130-098-803 RRID:AB_2660783), Anti-GLAST (ACSA-1)-((PE) Cat# 130-098-804 RRID:AB_2660782), Anti-CD15-((PE) Cat# 130-091-375 RRID:AB_871623), Anti-CD15-((APC) Cat# 130-091-371, RRID:AB_871621), Anti-A2B5-((FITC) the antibody was discontinued—corresponds to Cat# 130-093-394,

**TABLE 1** Antibody and MicroBeads list

| Antibody | Species | Clone | RRID |
|---|---|---|---|
| Anti-Agt | Rabbit | EPR22199-203 | |
| Anti-ACSA-2 | Rat | IH3-18A3 | RRID:AB_2651192 |
| Anti-ACSA-2-(FITC) | Rat | IH3-18A3 | RRID:AB_2727421 |
| Anti-ACSA-2-(APC) | Rat | IH3-18A3 | RRID:AB_2727361 |
| Anti-ACSA-2-(PE) | Rat | IH3-18A3 | RRID:AB_2727361 |
| Anti-ACSA-2 Microbeads | Rat | IH3-18A3 | |
| Anti-A2B5-(FITC) | Mouse | 105-HB2 | RRID:AB_1036062 |
| Anti-Biotin MicroBeads | | | RRID:AB_244365 |
| Anti-BLBP | Rabbit | | RRID:AB_10000325 |
| Anti-CD140a(PDGFRα)-(PE) | Rat | APA5 | RRID:AB_2655070 |
| Anti-CD140a(PDGFRα)-(APC) | Rat | APA5 | RRID:AB_2655072 |
| Anti-CD15-(PE) | Mouse | VIMC6 | RRID:AB_871623 |
| Anti-CD15-(APC) | Mouse | VIMC6 | RRID:AB_871621 |
| Anti-CD45 MicroBeads | Rat | 30F11 | |
| Anti-GFAP | Rabbit | | RRID:AB_10013382 |
| Anti-GFAP | Goat | | RRID:AB_641021 |
| Anti-GFP | Rabbit | | RRID:AB_221569 |
| Anti-GFP | Chicken | | RRID:AB_2307313 |
| Anti-GABA transporter 3 | Rabbit | | RRID:AB_90779 |
| Anti-GLAST | Mouse | ACSA-1 | RRID:AB_10829302 |
| Anti-GLAST | Guinea Pig | | RRID:AB_90949 |
| Anti-GLAST (ACSA-1)-(APC) | Mouse | ACSA-1 | RRID:AB_2660783 |
| Anti-GLAST (ACSA-1)-(PE) | Mouse | ACSA-1 | RRID:AB_2660782 |
| Anti-GLAST-Biotin | Mouse | ACSA-1 | RRID:AB_10829314 |
| Anti-Glutamate receptor 1 | Rabbit | | RRID:AB_2113602 |
| Anti-KI67 | Rabbit | | RRID:AB_442102 |
| Anti-L1CAM-PE | Rat | | RRID:AB_2655591 |
| Anti-MYBPC1 | Rabbit | | |
| Anti-NG2 | Rabbit | | RRID:AB_91789 |
| Anti-NPY | Rabbit | | RRID:AB_1566510 |
| Anti-OLIG2 | Rabbit | | RRID:AB_2299035 |
| Anti-PE MicroBeads | | | RRID:AB_244373 |
| Anti-PSA-NCAM MicroBeads | Mouse | 2-2B | |
| Anti-S100beta | Mouse | Clone SH-B1 | RRID:AB_477499 |
| Anti-Ter119 MicroBeads | Rat | Ter-119 | |

RRID:AB_1036062), Anti-CD140a(PDGFRα)-((PE) Cat# 130-109-783, RRID:AB_2655070), Anti-CD140a(PDGFRα)-((APC) Cat# 130-109-784, RRID:AB_2655072) (all Miltenyi Biotec; titer according to the manufacturer's protocol). For the gating, we used side scatter versus forward scatter to determine neural cells. Cell debris and dead cells were identified by Propidium Iodide (Miltenyi Biotec) and excluded from the analyses. After excluding doublets, the frequencies of stained cells were identified using the channel appropriate for the selected fluorophore (compare Figure 3). Data were acquired on a MACSQuant Analyzer 10 and processed using the MACSQuantify Software (both Miltenyi Biotec). Frequencies present the percentage of gated cells.

## 2.4 | Cell isolation by magnetic cell separation

Cerebella (either sex) derived from wild-type and β-actin-GFP mice were dissected, pooled, dissociated, and blocked with FcR Blocking Reagent, mouse (Miltenyi Biotec). The Anti-ACSA-2 MicroBead Kit allowed for efficient enrichment of ACSA-2$^+$/GLAST$^+$ cells to high purity using an MS column (both Miltenyi Biotec). The enrichment of ACSA-2$^-$/GLAST$^+$ cells required a two-step isolation protocol. First, single cells were labeled with Anti-L1CAM-PE (titer 1:2.5) (Cat# 130-102-865, RRID:AB_2655591) and then incubated with a mix of Anti-PE MicroBeads (1:2.5) Cat# 130-048-801, RRID:AB_244373), Anti-ACSA-2 MicroBeads (1:2.5), Anti-CD45 MicroBeads (1:10), Anti-PSA-NCAM MicroBeads (1:10), and Anti-Ter119 MicroBeads (all MicroBead conjugates from Miltenyi Biotec) and processed using an LD column (Miltenyi Biotec) to deplete the unwanted cell types. The negative fraction was then incubated with Anti-GLAST-Biotin (Cat# 130-095-815, RRID:AB_10829314; Miltenyi Biotec) and then with Anti-Biotin MicroBeads (Cat# 130-091-256, RRID:AB_244365; Miltenyi Biotec). Cells were subjected to magnetic cell separation to isolate GLAST positive cells.

## 2.5 | Homochronic and heterochronic transplantations *in vivo*

MACS sorted cells from P1–P3 β-actin-GFP donor mice (Okabe et al., 1997), of either sex, were grafted into the cerebellum of neonatal β-actin-GFP$^-$ littermates or P30 CD1 wild-type mice, of either sex (60.000 cells/μl, two injections of 1 μl each). The vermis was targeted according to previously established protocols (neonatal: from lambda −1 mm AP, ±0.5 mm ML, 1 mm DV; adult: from bregma −7.2 mm AP, ±0.8 mm ML, 1.5–2 mm DV) (Carletti et al., 2008; Leto et al., 2006). Mice were sacrificed 30 days post-transplantation. Donor cells in the host tissue were recognized by intrinsic GFP expression. Their phenotypic traits were investigated by morphometric evaluation, layering, and expression of cell type-specific markers.

## 2.6 | Immunohistology

Adult and neonatal mice, of either sex, were anesthetized with ketamine (100 mg/kg body weight, Ketavet, Bayer) and xylazine (5 mg/kg body weight, Rompun, Bayer) and perfused with 4% paraformaldehyde (PFA) in 0.12 M phosphate buffer (PB). Brains were postfixed in 4% PFA overnight (o/n), stored in PBS and cryoprotected using 30% sucrose in 0.12 M PB. Tissue was embedded in OCT (TissueTEK), sectioned in 30-μm-parasagittal plane using a cryostat and collected as floating sections in PBS or placed directly on glass slides. For vibratome sectioning (adult brains), brains were embedded in 3.5% agarose plus 8% sucrose (both Merck), sectioned into 40–60 μm slices and collected as floating sections.

Immunostainings for ACSA-2 were optimized for early postnatal tissue. A periodate-lysine-paraformaldehyde treatment (PL-PFA: 0.01 M NaIO$_4$, 0.1 M lysine, 4% PAF in 0.15 M NaP) was used to preserve glycoprotein antigens (McLean & Nakane, 1974). For immunofluorescence staining, sections were blocked with 1.5%–4% goat or donkey serum depending on the species of the secondary antibody (1 hr RT; Dianova; Jackson ImmunoResearch). Sections were incubated in PBS with serum and 0%–0.3% Triton-X-100 (Sigma) o/n at 4°C or RT with following antibodies (compare also RRID—Table 1): Anti-ACSA-2 (1:50 (neonatal)—1:250 (adult), (Miltenyi Biotec; Cat# 130-099-138, RRID:AB_2651192), Anti-AGT (1:200 (rabbit monoclonal [EPR22199-203] Cat#ab229005)), Anti-BLBP (1:500; rabbit, polyclonal; Millipore Cat# ABN14, RRID:AB_10000325), Anti-NG2 (1:200; rabbit polyclonal; Millipore Cat# AB5320, RRID:AB_91789), Anti-GFP (1:700, rabbit, polyclonal; Molecular Probes Cat# A-11122, RRID:AB_221569), Anti-GFP (1:700, Aves Labs Cat# GFP-1010, RRID:AB_2307313), Anti-GLAST (ACSA-1) (1:50-1:250, Miltenyi Biotec Cat#130-095-822, RRID:AB_10829302), Anti-GLAST (1:500, Millipore Cat# AB1783, RRID:AB_90949), Anti-GFAP (1:1,000, Agilent Cat# Z0334, RRID:AB_10013382), Anti-GFAP (1:250, Santa Cruz Biotechnology Cat# sc-6170, RRID:AB_641021), Anti-Glutamate receptor 1 (1:250, Millipore Cat# AB1504, RRID:AB_2113602), Anti-Ki67 (1:500, Leica Biosystems Cat# NCL-Ki67p, RRID:AB_442102), Anti-MYBPC1 (1:500, Sigma-Aldrich Cat#SAB3501005), Anti-NPY (1:200, Abcam Cat# ab30914, RRID:AB_1566510), Anti-PARVALBUMIN (1:1,500, mIgG1; Swant Cat# 235, RRID:AB_10000343), Anti-S100β (1:1,000, Sigma-Aldrich Cat# S2532, RRID:AB_477499), Anti-OLIG2 (1:500, Millipore Cat# AB15328, RRID:AB_2299035). For MYBPC1 and NPY detection, the antigens were recovered at 80°C for 20 min in sodium citrate solution (10 mM, pH 6), while for AGT detection, a Tris-EDTA Buffer (10 mM Tris Base, 1 mM EDTA, 0.05% Tween 20, pH 9) was used. After washing, the secondary species-specific antibodies conjugated with Alexa Fluor 488, Alexa Fluor 555, Alexa Fluor 594, Alexa Fluor 647 (all 1:500; Invitrogen) or Cy3 (1:500; Jackson ImmunoResearch) were added for 1 hr (RT). In some cases, an anti-mouse biotinylated secondary antibody (1:500; Jackson ImmunoResearch) was used followed by 1 hr incubation at RT with

**TABLE 2** Statistical parameters of the pairwise comparison of $A^+/G^+$ and $A^-/G^+$ frequencies in different layers of the postnatal (P3) cerebellum

| Comparison | Main effect | Layer | N (animals) | Applied test | Sidak's multiple comparison (p) |
|---|---|---|---|---|---|
| *Over total (DAPI+) cells* | | | | | |
| $A^-/G^+$ versus $A^+/G^+$ | Interaction: $F(2, 12) = 74.08$; $p < 0.0001$ (***) | PCL/ML | $n = 3$ | Two-way ANOVA | <0.0001 (***) |
| | Layer: $F(2, 12) = 45.58$; $p < 0.0001$ (***) | GL | | | 0.0259 (*) |
| | Cell type: $F(1, 12) = 127.7$; $p < 0.0001$ (***) | PWM | | | 0.9989 |
| *Over total BLBP$^+$ astroglial-like progenitors* | | | | | |
| $A^-/G^+$ versus $A^+/G^+$ | Interaction: $F(2, 12) = 453.8$; $p < 0.0001$ (***) | PCL/ML | $n = 3$ | Two-way ANOVA | <0.0001 (***) |
| | Layer: $F(2, 12) = 0.1286$; $p = 0.8806$ (***) | GL | | | <0.0001 (***) |
| | Cell type: $F(1, 12) = 1,958$; $p < 0.0001$ (***) | PWM | | | 0.7386 |

**TABLE 3** Statistical parameters of the pairwise comparison of $A^+/G^+$ and $A^-/G^+$ frequencies during different stages of early postnatal cerebellar development, analyzed by flow cytometry

| Age | N (animals) | Applied test | t | Main effect (p) |
|---|---|---|---|---|
| P1 | $n = 7$ (per condition) | Paired $t$ test | 4.261 | 0.008 (**) |
| P3 | $n = 7$ (per condition) | Paired $t$ test | 4.197 | 0.0057 (**) |
| P5 | $n = 9$ (per condition) | Paired $t$ test | 5.345 | 0.0007 (***) |
| P7 | $n = 7$ ($A^+/G^+$); $n = 9$ ($A^-/G^+$) | Paired $t$ test | 6.92 | 0.0005 (***) |

**TABLE 4** Statistical parameters of the pairwise comparison of glial markers on A+/G+ and $A^-/G^+$ cells in the developing cerebellum, analyzed by flow cytometry

| Age | N (animals) | Applied test | t | Main effect (p) |
|---|---|---|---|---|
| P2 | $n = 3$ (per condition) | Unpaired $t$ test | 11.23 | 0.0054 (**) |
| P6 | $n = 3$ (per condition) | Unpaired $t$ test | 5.788 | 0.0139 (*) |
| P3 | $n = 3$ (per condition) | Unpaired $t$ test | 5.037 | 0.032 (*) |

Streptavidin-421 (1:200; BioLegend). Cell nuclei were stained with 4′, 6-diamidino-2-phenylindole (DAPI; Fluka) or DRAQ5™ Alexa Fluor 647 (5 µM Abcam). Cells and sections were mounted with Mowiol or Mounting medium (Dako). For histological quantifications at least three sections per animals were analyzed. The absolute number of animals used is given in Tables 2–5.

## 2.7 | Imaging

Histological analyses of the grafts were performed using an E-800 Nikon microscope. Confocal images were taken on a Zeiss (LSM) or a Leica (TCS SP-5) confocal microscope. Images were processed using NIH ImageJ (RRID:SCR_003070) software.

## 2.8 | Gene expression profiling.

Biological triplicates of $A^-/G^+$ and $A^+/G^+$ cells, derived from either sex, were isolated by magnetic cell separation (neonatal cerebellum P0/P1—50.000 cells/sample) and shock frozen. RNA was isolated with the RNeasy Mini Kit (Qiagen) and quality check was performed on the Agilent 2100 Bioanalyzer platform. Total RNA (up to 50 ng) was amplified and Cy3-labeled according to the manufacturer's protocol (Agilent Low Input Quick Amp Labeling Kit). Dye incorporation and yield of cRNA were measured on the NanoDrop. Six hundred nanogram Cy3-labeled fragmented cRNA were hybridized overnight (17 hr, 65°C) to Agilent Whole Mouse Genome Oligo Microarrays $8 \times 60$ K. Data were quantile normalized and differentially expressed probes were identified by a combination of effect size and statistical significance. Only probes with at least fourfold median up- or downregulation were compared. Out of these, only samples with a Benjamini–Hochberg corrected $p$ value (two-tailed Student's $t$ test) less or equal to 0.05 were considered relevant. Heatmaps of median centered log2 intensities were generated by using the Multi-Experiment Viewer Software (MeV v4.9) (RRID:SCR_001915). The volcano plot was generated with the statistical software R (v3.6.0) (RRID:SCR_001905).

The data discussed and methods used in this publication have been deposited in NCBI's Gene Expression Omnibus (RRID:SCR_005012) and are accessible through GEO Series accession number GSE117886.

**TABLE 5** Test of independence for the cell populations revealed upon grafting

| N (animals) | Applied test | z | Main effect (p) |
|---|---|---|---|
| BG versus astrocytes: $A^+/G^+$ ($n = 7$); $A^-/G^+$ ($n = 9$) | Chi-squared test | 21.98 | <0.0001 (****) |
| Glia versus interneurons: $A^+/G^+$ ($n = 7$); $A^-/G^+$ ($n = 9$) | Chi-squared test | 19.08 | <0.0001 (****) |

## 2.9 | Analysis of sc/snRNA-seq published data sets

To better understand the nature of the $A^+/G^+$ and $A^-/G^+$ cells and identify candidate marker genes for these two populations, we analyzed two recently published data sets of sc/snRNA-seq in the embryonic/postnatal and adult mouse cerebellum (Kozareva et al., 2020; Vladoiu et al., 2019). The sequence data of these two data sets are available in the Gene Expression Omnibus (Boulay et al., 2017) repository under accession GSE118068 and at the Neuroscience Multi-omics (NeMO) Archive (https://singlecell. broadinstitute.org/single_cell/study/SCP795/a-transcriptomic-atlas -of-the-mouse-cerebellum), respectively. For both data sets, the RNA-seq count matrices were downloaded and processed with the Seurat R package (v 3.6.2).

Specifically, for the first data set (Vladoiu et al., 2019), we focused on the data obtained at P0 and P7. scRNAseq data were subjected to quality control, filtering of low-quality cells, log-normalization, scaling, and linear dimensional reduction (PCA) with the top 20 principal components. Afterwards, a graph-based clustering approach was used to identify, for each developmental stage, the astrocyte population based on known marker genes (*Slc1a3*, *Fabp7*, *Aldh1l1*, and *S100b*). The count matrix of the astrocyte population was therefore extrapolated and further subjected to the same clustering approach to identify distinct subpopulations, as described in Figures S1–S3 (Cerrato et al., 2018; Farmer et al., 2016; Salvi et al., 2019). Specifically, the enriched expression of *Gdf10*, *Gria1*, and *Gria4* allowed identifying the BG/BG progenitor subpopulations, while other parenchymal astrocytes/astrocyte progenitors were classified as those cells showing higher expression of *Aqp4*. The expression of *Ptf1a*, *Dcx*, and *Ascl1*, typically high in neuronal progenitors (Gleeson et al., 1999; Grimaldi et al., 2009; Hoshino et al., 2005) allowed the classification of one of the clusters observed at P0 as a neurogenic progenitors cluster.

For the adult data set (Kozareva et al., 2020), the original classification of the distinct cerebellar cell types provided by the authors (Figure 4) was used to identify the astrocyte population within the data set. The count matrix of this population was then extracted and the data obtained from the distinct biological replicates ($n = 6$) were integrated according to the SCTransform workflow in Seurat (Hafemeister & Satija, 2019), to correct for batch effects. Subsequently, cells were subjected to clustering to identify the distinct astrocyte subpopulations (Figures S1–S3).

The Seurat FindAllMarkers function was applied for both data sets to identify the differentially expressed genes of each cluster, selecting only significantly upregulated genes ($p < 0.01$) at least 1.4-fold overexpressed in more than 25% of the cells belonging to the subpopulation of interest (when compared to all other astroglial cells in the data set under analysis). For the different developmental stages, the lists of the top 100 differentially expressed genes of the clusters classified as BG/BG progenitors, parenchymal astrocytes/astrocyte progenitors, or neurogenic progenitors were used.

## 2.10 | Statistics

Statistics were calculated with GraphPad Prism (GraphPad Software Inc., California USA) (RRID:SCR_002798). Error bars in the graphs represent average values $\pm$ standard deviation (*SD*).

## 3 | RESULTS

### 3.1 | ACSA-2 is expressed by subpopulations of astroglial cells in the adult and developing cerebellum

As others and we demonstrated previously, the ACSA-2 reveals a broad expression in the murine adult cerebellum. Initial studies with the general astrocyte marker GLAST disclosed a non-overlapping pattern in distinct astroglial populations of the adult cerebellum (Kantzer et al., 2017). To address this in more detail, high-resolution analyses were performed and the dynamic and localization of ACSA-$2^+$ astrocytes were investigated using brain lipid-binding protein (BLBP), besides GLAST; two markers that are known to be expressed by cerebellar astrocytes (Anthony et al., 2004; Cerrato et al., 2018). In line with former studies (Holmseth et al., 2012; Jungblut et al., 2012; Storck et al., 1992), GLAST was broadly expressed by BG, an astrocyte cell type closely associated with Purkinje neurons in the ML (Figure 1a″,c″; arrowheads in a). BG, that also present BLBP labeling (Figure 1b″; arrowheads in b), show weak expression of ACSA-2 in the ML (Figure 1a′–c′; arrowheads in a,b). ACSA-2 was predominantly found in the GL, just below the PCL but not in the PCL itself (Figure 1a′). Remarkably, ACSA-2 never appeared as prominent on BG fibers (Figure 1b′,c′; arrowheads in b) as commonly seen for BG markers such as GLAST or BLBP (Figure 1b″,c″; arrowheads in b). In contrast to the ML, the intensities of GLAST and BLBP were lower in velate astrocytes (Figure 1a″–c″). In turn, the latter compartment comprised astrocytes displaying strong ACSA-2 expression (Figure 1a′–c′). The inner part of the cerebellum, the WM, revealed co-staining of ACSA-2 and GLAST (Figure 1c).

As cerebellar astrocytes are mainly generated postnatally (Cerrato et al., 2018), we addressed the ontogenesis of ACSA-$2^+$ astrocytes early after birth using immunostainings (Figure 2). To assess

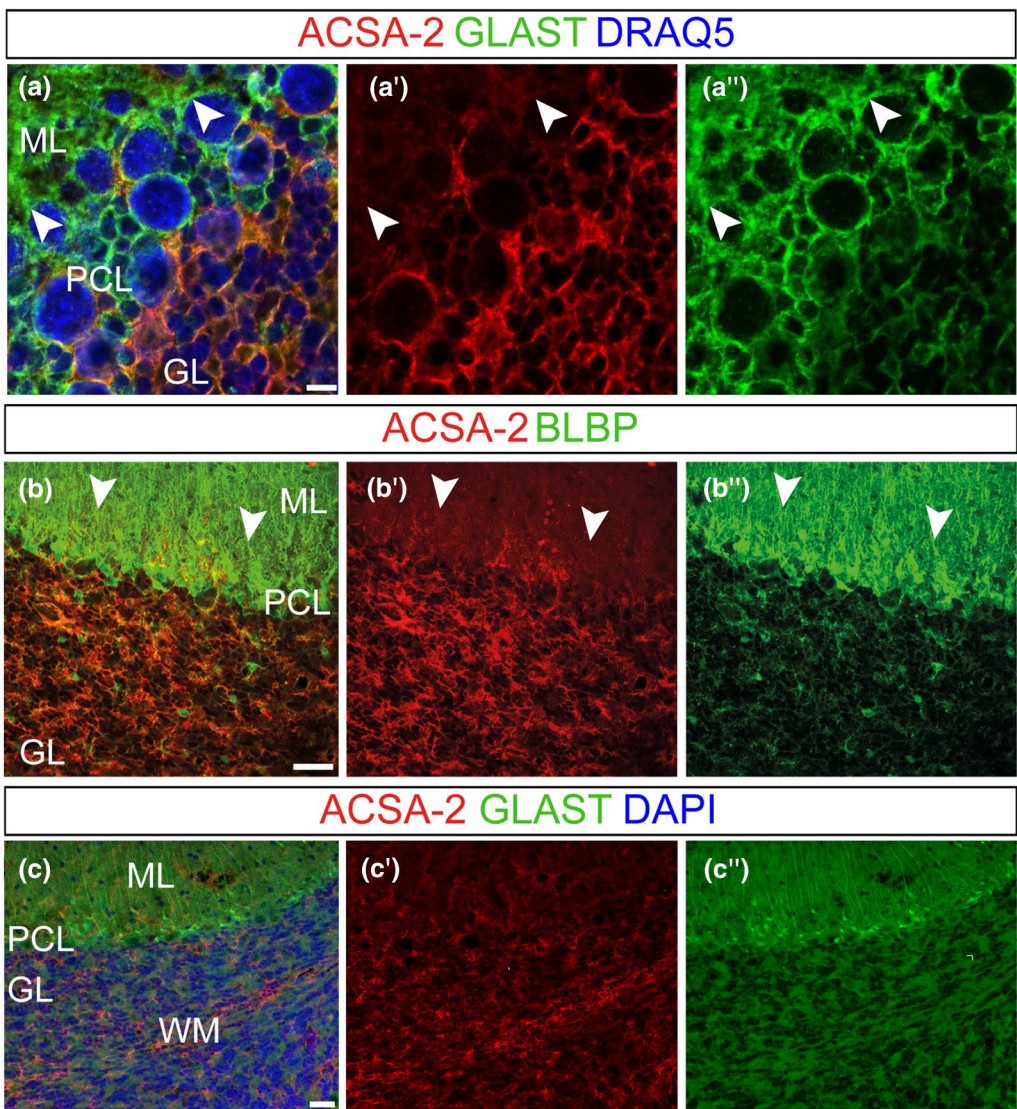

**FIGURE 1** ACSA-2 presents a unique expression pattern in the adult cerebellum. Coronal (a) and sagittal (b,c) adult mouse (younger than 2 months) cerebellar sections were used to investigate ACSA-2 expression. Confocal stacks identified ACSA-2, unlike the common astrocyte markers GLAST (a,c) and BLBP (b), to be distinct in the three cerebellar cortical layers (a,c). GLAST⁺ BLBP⁺ BG processes in the ML (a″,b″,c″; arrowheads in a″,b″) revealed weak ACSA-2 expression (a′–c′; arrowheads in a′,b′). High ACSA-2 expression was found on velate protoplasmic astrocytes in the GL (a′–c′). ACSA-2 is further expressed by a third type of cerebellar glia, the fibrous astrocytes in the WM (c). Nuclear stain: DRAQ5 (a), DAPI (c). Scale bars: 5 μm (a), 50 μm (b) 30 μm (c). GL, granular cell layer; ML, molecular layer; PCL, Purkinje cell layer; WM, white matter [Color figure can be viewed at wileyonlinelibrary.com]

the specificity of ACSA-2 labeling to astroglial cells, we co-stained ACSA-2 with the neural/glial antigen (NG2), which is expressed by oligodendrocyte precursor cells (OPCs) (Polito & Reynolds, 2005) (Figure S4a). No co-expression of the two markers was detected (Figure S4a) and this result was further confirmed by cytofluorimetric assessment of positivity for ACSA-2 and the other OPC marker platelet-derived growth factor receptor alpha (PDGFRα) (Figure S4b). Thus, we concluded that ACSA-2 only marks the astroglial lineage. Quantification of BLBP⁺ astroglial progenitors at P3 showed that they are present throughout the cerebellar layers: they represent a small fraction of cells in the PWM and GL while they are the major population in the PCL/ML (Figure S4c). Here, bona fide BG progenitors in the PCL and GL astrocytes, marked by BLBP

(Figure 2a″,c″) and GLAST (Figure 2c″′), were ACSA-2⁻ (Figures 2a′,c′ and S4d). ACSA-2⁺ cells were exclusively found in discrete cell clusters of the PWM, in the depth of the cerebellar parenchyma and at the basis of lobules (Figure 2a,a′,b,b′,c,c′). Co-expression with BLBP and GLAST confirmed these ACSA-2⁺ cells to be of astroglial lineage (Figure 2b,c). However, only half of BLBP⁺ and GLAST⁺ astroglial precursors expressed ACSA-2 (Figures 2a–c and S4c,d. PCL/ML $p < 0.0001$ over DAPI and BLBP-positive cells; GL $p = 0.0259$ over DAPI and $p < 0.0001$ over BLBP, two-way ANOVA with Sidak's multiple comparison; $n = 3$; Table 2). Investigating the PWM further, we detected ACSA-2⁺/GLAST⁺ (Figure 2d–d″″, white arrowheads) as well as ACSA-2⁻/GLAST⁺ cells (Figure 2d–d″″, filled white arrows) and a very rare population of ACSA-2⁺/GLAST⁻ cells (not included in

the quantifications, Figure 2d–d″″, empty arrows). These observations suggested that ACSA-2 and GLAST define two major astroglial cell subsets in the developing cerebellum: ACSA-2⁺ expressing cells that co-express GLAST⁺ (A⁺/G⁺) and cells that are devoid of ACSA-2 but express GLAST (A⁻/G⁺). The occurrence of these subsets was later confirmed by flow cytometry (Figure 3).

Since the PWM hosts proliferating progenitors with astroglial traits as well as post-mitotic astrocytes, we inquired whether ACSA-2⁺ astroglial cells could represent early post-mitotic elements (Figure S5a–d: close up in b and orthogonal projection in c). Co-labeling with the proliferation marker Ki67 showed that at P3 8.4% ± 1.55% (Figure S5d) of the ACSA-2⁺ cells in the PWM proliferate (Figure S5a and close

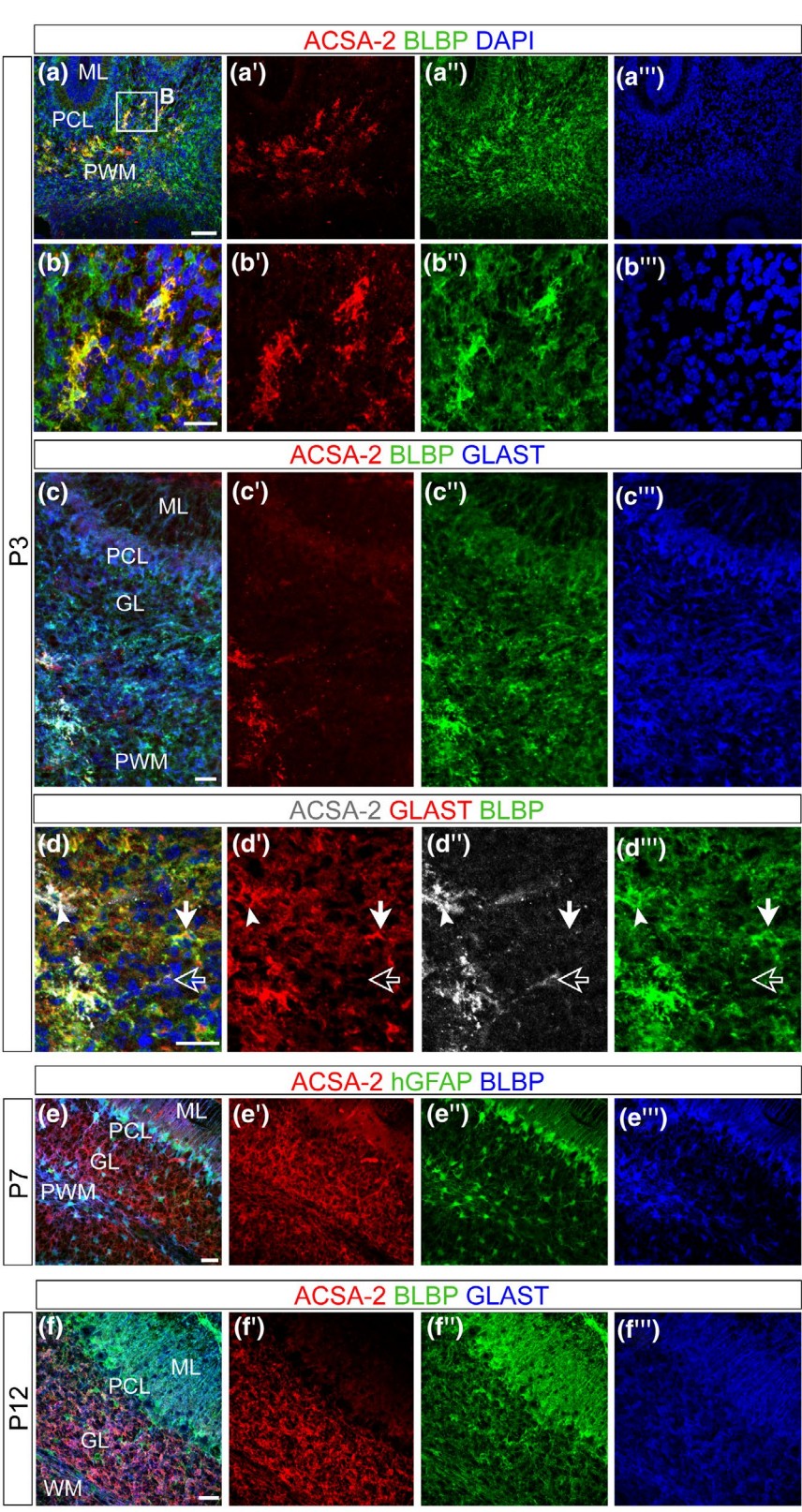

**FIGURE 2** ACSA-2 defines a subpopulation of astrocyte progenitors in the neonatal cerebellum. The ontogeny of ACSA-2⁺ cells was investigated using confocal stacks of P3 (a–d), P7 (e) and P12 (f) sagittal sections. At P3, ACSA-2 was not detected in the nascent ML (a′,c′) and neither in the PCL (a′,c′). BG progenitors, which are marked by BLBP and GLAST (a″,c″,c‴), are not expressing ACSA-2. ACSA-2 is present in the PWM (a′,b′,c′,d″) but it is not as broadly expressed as the common markers GLAST and BLBP (a″,c″,c‴,d′,d‴). Within the GLAST⁺ population (of the PWM) ACSA-2 positive (white arrowheads in d) as well as ACSA-2 negative cells (filled white arrows in d) and a very rare population of ACSA-2⁺/GLAST⁻ cells (empty arrows in d) were detected (d–d″). At P7, the labeling of ACSA-2 appeared in the PWM and the GL and overlapped with the reporter expression of the human glial fibrillary acidic protein (hGFAP) in the PWM and the GL (e). At P12, ACSA-2 expression has expanded to include BLBP⁺ and GLAST⁺ cells in both inner compartments: the GL and the WM (f). DAPI, nuclear stain. Scale bars: 100 μm (a), 30 μm (c,e,f), 50 μm (b,d). GL, granular cell layer; ML, molecular layer; PCL, Purkinje cell layer; PWM, prospective white matter; WM, white matter [Color figure can be viewed at wileyonlinelibrary.com]

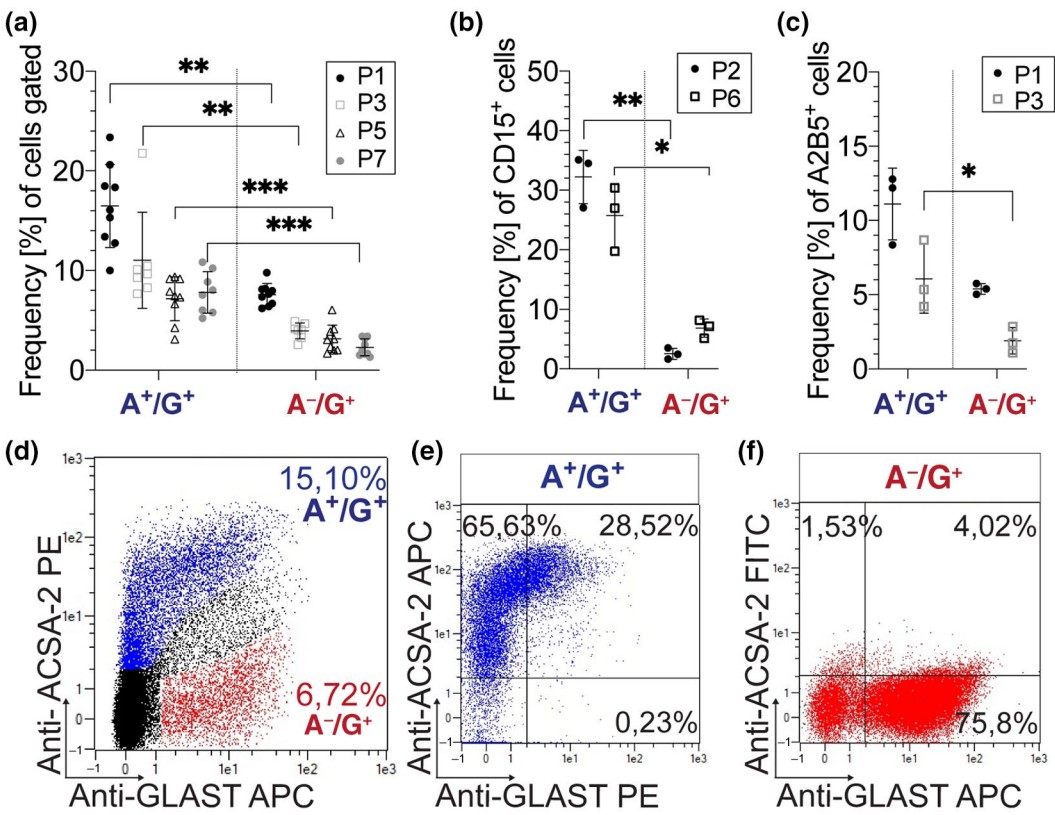

**FIGURE 3** Flow cytometry reveals a gliogenic marker expression on A⁺/G⁺ cells. (a–d) Cerebella (P1–P7) were dissociated, stained, and analyzed by flow cytometry. (d) Anti-ACSA-2 and Anti-GLAST show a highly dynamic range of dim and broad expression. The marker analyses were based on the presence (A⁺/G⁺ as marked in blue in d) and absence of ACSA-2 (A⁻/G⁺ as shown in red in d). (a) Frequencies of A⁺/G⁺ cells were significantly higher than A⁻/G⁺ cells ((P1): **$p = 0.008$; (P3): **$p = 0.0057$; (P5): ***$p = 0.0007$; (P7): ***$p = 0.0005$; paired $t$ test; $n = 7$). At early neonatal age (P2/P3), the glial precursor markers CD15 (b) and A2B5 (c) were significantly higher expressed on A⁺/G⁺ cells compared to A⁻/G⁺ cells. The difference in CD15 expression was maintained at P6 (b: (P1): **$p = 0.0054$; (P6): *$p = 0.0139$; unpaired $t$ test; $n = 3$) (c: (P3): *$p = 0.032$; unpaired $t$ test; $n = 3$). (e,f) Magnetic isolation allowed for efficient enrichment of A⁺/G⁺ (86.34% ± 9.3% $n = 3$) (e) and A⁻/G⁺ cells (76.26% ± 2.179% $n = 11$) (f) from dissociated cerebellar tissue (P1–P3) [Color figure can be viewed at wileyonlinelibrary.com]

up in b). This frequency is comparable to that of the whole astroglial-like progenitor population in the PWM (8.9% ± 0.3%, Parmigiani et al., 2015). This finding indicates that ACSA-2 does not exclusively identify post-mitotic elements but does include proliferating progenitors to the same extent of the rest of the astroglial population.

We then investigated a second developmental stage (P7) using a transgenic mouse-line, that expresses GFP under the control of the human glial fibrillary acidic protein (hGFAP) promoter (Figure 2e). The reporter was, like BLBP, broadly expressed throughout the layers of the developing cerebellum (Figure 2e″,e‴). In contrast to the pattern

seen at P3, at P7, ACSA-2 expression was found in the PWM and in areas of the GL (Figure 2e′). At P12, before the completion of cerebellar maturation, ACSA-2 already displayed an expression pattern like that seen in the adult: it decorated velate protoplasmic astrocytes of the GL (Figure 2f′). Still, the developing BG, co-expressing GLAST and BLBP (Figure 2f″,f‴), displayed low levels of ACSA-2.

In summary, ACSA-2 expression starts during the first postnatal days in astroglial-like cells in the PWM. It then gradually expands throughout the cerebellum to label astrocytes in the GL and WM, respectively, and to a lesser extent BG in the ML.

## 3.2 | ACSA-2 and GLAST protein levels define two cell populations in the neonatal cerebellum

Expanding further on the nature of ACSA-2$^+$ cells, we set out to check the expression of ACSA-2 and GLAST in the developing cerebellum

at the single cell level. At P1, the flow cytometry analysis revealed the presence of both proteins (Figure 3d). The co-labeling of ACSA-2 and GLAST appeared very dynamic with dim and bright levels of both markers. We decided to focus here on the populations that are most discriminative in ACSA-2 expression: ACSA-2$^+$/GLAST$^+$ (A$^+$/G$^+$)

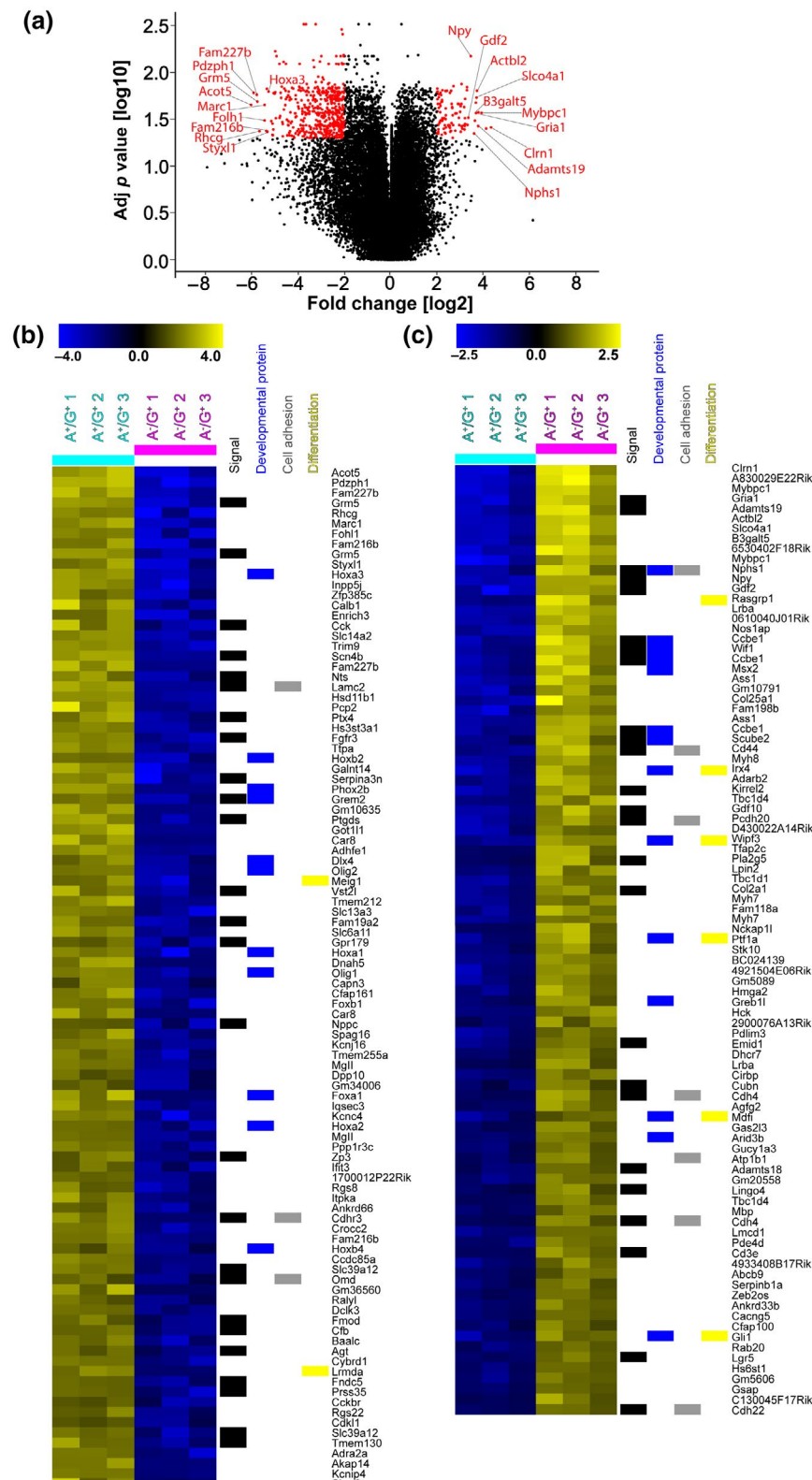

**FIGURE 4** Transcriptional analysis identifies major differences between A⁻/G⁺ and A⁺/G⁺ precursors. (a) Volcano plot of microarray-based gene expression data. Probes with an at least fourfold median up- or downregulation, for both comparisons, and a Benjamini–Hochberg corrected $p$ value (two-tailed Student's $t$ test) of less than or equal to 0.05 are highlighted in red. The top 10 genes showing the highest upregulation in the A⁺/G⁺ (left) or A⁻/G⁺ (right) fraction are labeled. (b) Heatmap depicts the top 100 hits of the microarray probes found in the A⁺/G⁺ fraction (light blue header bar) compared to the A⁻/G⁺ fraction (magenta header bar). All probes show at least fourfold upregulation and present a Benjamini–Hochberg corrected $p$ value (two-tailed Student's $t$ test) of less than or equal to 0.05. Probes without gene assignments were excluded. Assignments of the microarray probes to the Uniprot keywords "Signal," "Developmental protein," "Cell adhesion," and "Differentiation" are indicated on the right hand side of the heatmap. (c) Heatmap depicts the top 93 median-centered log2 values of microarray probes with an at least fourfold median upregulation in the A⁻/G⁺ fraction (magenta header bar) compared to the A⁺/G⁺ fraction (light blue header bar) and with a Benjamini–Hochberg corrected $p$ value (two-tailed Student's $t$ test) of less than or equal to 0.05. Probes without gene assignments were excluded. Assignments of the microarray probes to the Uniprot keywords "Signal," "Developmental protein," "Cell adhesion," and "Differentiation" are indicated on the right hand side of the heatmap [Color figure can be viewed at wileyonlinelibrary.com]

cells and ACSA-2⁻/GLAST⁺ (A⁻/G⁺) cells. ACSA-2⁺/GLAST⁻ cells were considered together with the A⁺/G⁺ population since this subset was rarely detected *in vivo* (Figure 2d).

At the early postnatal stage, GLAST is broadly and homogeneously expressed outside the PCL while ACSA-2 is restricted to the PWM (Figure 2). The flow analyses using dissociated tissue revealed a significantly higher frequency of A⁺/G⁺ cells than A⁻/G⁺ cells when comparing different stages of early postnatal development (Figure 3a). We then characterized the A⁺/G⁺ and A⁻/G⁺ population with known glial markers. The Lewis X antigen (CD15) has been proposed as a marker for astrogliogenic committed precursors in the cerebellum (Figure 3b) (Fleming et al., 2013). CD15 expression appeared more prominent on A⁺/G⁺ cells (Figure 3b) compared to the A⁻/G⁺ population (P2: ***$p$ = 0.0054; P6: **$p$ = 0.0139; $n$ = 3; paired $t$ test; Table 3). We also tested the ganglioside marker A2B5 (Fredman et al., 1984; Lee et al., 2000). At P3, A2B5 was significantly higher on A⁺/G⁺ compared to A⁻/G⁺ (Figure 3c) (*$p$ = 0.032; paired $t$ test; $n$ = 3; Table 4). In conclusion, these data show that markers of gliogenic progenitors (A2B5 and CD15) are particularly enriched in A⁺/G⁺ cells.

To perform comparative analyses on separated cell fractions we established magnetic isolation protocols (see Methods). Using these protocols, viable A⁺/G⁺ cells (Figure 3e) and A⁻/G⁺ cells (Figure 3f) were enriched. Of note, the frequency of granule neurons detected by L1CAM (Inaguma et al., 2016) showed frequencies of 35.4% ± 6.2% ($n$ = 5) in the single cell suspensions. Upon depletion, the residual amount of L1CAM⁺ cells in the A⁻/G⁺ population was under 1% (0.93% ± 0.64% ($n$ = 3)).

### 3.3 | Transcriptomic analysis shows distinct astrocyte features in A⁻/G⁺ and A⁺/G⁺ precursors

To further understand the role of ACSA-2 on GLAST⁺ precursors we analyzed their gene expression profile. Independent triplicates of A⁻/G⁺ and A⁺/G⁺ samples were enriched from neonatal cerebella (P0 and P1) and the RNA was subjected to whole mouse genome microarrays. Only probes that presented at least fourfold median up- or downregulation and a Benjamini–Hochberg corrected $p$ value

(two-tailed Student's $t$ test) of less than or equal to 0.05 were considered relevant. Of note, the expression of *Atp1b2*, the target of the anti-ACSA-2 antibody, was four times higher in the A⁺/G⁺ fraction than in the A⁻/G⁺ fraction. The volcano plot (Figure 4a) presents an overview of the comparative expression profile. It highlights the top 10 hits, the genes with the highest differential expression, identified for the A⁺/G⁺ (Figure 4a, left side) and the A⁻/G⁺ sample set (Figure 4a, right side).

Among the top A⁺/G⁺ 10 hits, most were indeed reported in astrocytes or astroglial precursors (Boulay et al., 2017; Choi et al., 2014; Panatier & Robitaille, 2016; Pinto et al., 2018; Raikwar et al., 2019; Tomar et al., 2019; Umpierre et al., 2019; Wang et al., 2018; Zeisel et al., 2018), although no specific expression in cerebellar astroglial cells has been shown so far. For instance, metabotropic glutamate receptor 5 (*Grm5*), is implicated in astrocyte regulation of synaptic transmission and glutamate uptake (Panatier & Robitaille, 2016; Umpierre et al., 2019), astrocytic glutamate carboxypeptidase 2 (*Folh1*) is also known to modulate neurotransmission (Choi et al., 2014) and *Fam216b* was associated with astrocyte endfeet (Boulay et al., 2017; Pinto et al., 2018). The homeobox A3 protein (*Hoxa3*), was instead reported to be expressed in the rhombencephalon during hindbrain development (Zeisel et al., 2018).

The top A⁻/G⁺ 10 hits included, instead, three markers, namely myosin-binding protein C (*Mybpc1*), neuropeptide Y gene (*Npy*), and the glutamate ionotropic receptor AMPA type subunit 1 (*Gria*), previously described in BG or BG subsets (Kozareva et al., 2020; Reeber et al., 2018; Rodriques et al., 2019; Saab et al., 2012; Zeisel et al., 2018). Of note, also the transporter *Slco4a1*, actin beta-like protein (*Actbl2*), and clarin (*Clrn1*) have been reported in this astroglial type (Zeisel et al., 2018), while nephrin (*Nphs1*) was shown to be expressed by radial glia cells of the neonatal cerebellum (Putaala et al., 2001). Beta-1,3-galactosyltransferase 5 (*B3galt5*) is specifically expressed in certain lobules of the cerebellum (Rodriques et al., 2019), while no known association with astroglial cells or with a cerebellar origin was found for the growth/differentiation factor 2 (*Gdf2*)—also known as bone morphogenic protein 9 (*BMP9*)—and for the disintegrin and metalloproteinase with thrombospondin motifs 19 (*Adamts19*). Furthermore, within the top 50 differentially expressed genes in the A⁻/G⁺ population we found *Ptf1a*, a bHLH

transcriptional gene that defines cerebellar GABAergic interneurons (Hoshino et al., 2005), known to derive from PWM astroglial-like progenitors (Parmigiani et al., 2015; Vladoiu et al., 2019).

To obtain further functional information from the cell transcriptomes we expanded the differentially expressed gene list to 100 genes for the A⁺/G⁺ sample and to 93 for A⁻/G⁺ cells (Figure 4b,c, see Methods) and used these lists as input files for the annotation enrichment tool DAVID (da Huang et al., 2009a, 2009b). For both lists the gene set enrichment revealed significantly augmented Uniprot keywords as Developmental protein, Differentiation, Signal, and Cell adhesion, highlighting the precursor properties of those cells.

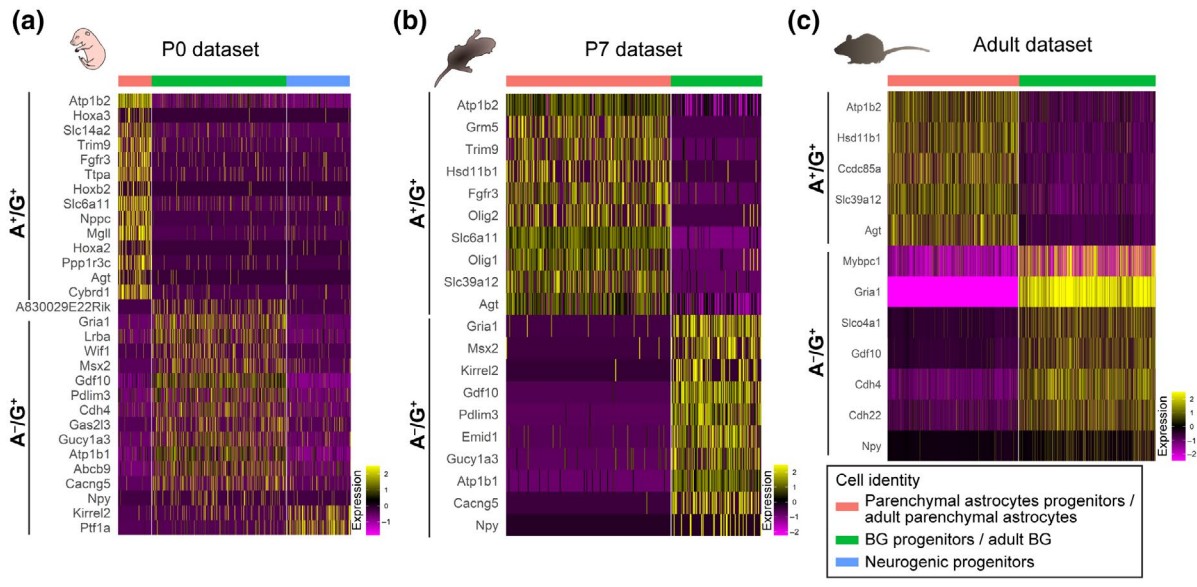

**FIGURE 5** Comparisons with available sc/snRNA-seq cerebellar data sets clarifies the nature of A$^+$/G$^+$ and A$^-$/G$^+$ precursors. Heatmaps show the average gene expression for *Atp1b2*, *Npy* and genes found to be shared by A$^+$/G$^+$ or A$^-$/G$^+$ cells and distinct subpopulations of astrocytes/astrocyte progenitors at P0 (a), P7 (b) or P60 (Adult, c) (values were obtained from the sc/snRNA-seq data sets). A$^-$/G$^+$ cells shared genes with the BG subpopulations and with the P0 neurogenic subpopulation, while A$^+$/G$^+$ cells showed genes in common with the sole parenchymal astrocyte subpopulation at all stages analyzed. P3 cerebellar sections were stained with candidate markers specific for A$^-$/G$^+$ (d–g) and A$^+$/G$^+$ (h,i) cells. GRIA1 (d), NPY (e), and MYBPC1 (f) were confirmed to be enriched in A$^-$/G$^+$ BG progenitors in the PCL and in the PWM (arrow in g) compared to A$^+$/G$^+$ cells (arrowheads in g), while AGT (h) and SLC6A11 (i) were specifically expressed in A$^+$/G$^+$ progenitors in the PWM. The astroglial identity was validated using an intrinsic GFP expressed under the control of the hGFAP promoter (d–g,i) or by GFAP staining (h) Scale bars: 20 µm. BG, Bergmann glia; GL, granular cell layer; ML, molecular layer; PCL, Purkinje cell layer; PWM, prospective white matter [Color figure can be viewed at wileyonlinelibrary.com]

Overall, the transcriptomic analysis revealed unexpected differences in the gene expression profile of A$^-$/G$^+$ progenitors, displaying markers of specific astroglia subtypes (BG and PWM precursors), and A$^+$/G$^+$ progenitors that instead exhibit more generic astroglial traits.

## 3.4 | A$^-$/G$^+$ and A$^+$/G$^+$ populations comprise distinct cerebellar astroglial precursor subsets associated with different lineages

To gain a better understanding of the identity of A$^-$/G$^+$ and A$^+$/G$^+$ precursors in the context of cerebellar development and cell type differentiation, we compared their gene expression profiles to those of cerebellar astroglial cells at two different developmental stages (i.e., P0 (Figure S1), and P7 (Figure S2)). Data were extracted from a recently published single-cell (sc) RNA-seq data set of the developing mouse cerebellum (Vladoiu et al., 2019). Namely, at both stages, we first identified and extrapolated the cells with astroglial features based on the expression of genes for typical astrocyte markers such as *Slc1a3*, *Fabp7*, *Aldh1l1*, and *S100b* (see Methods), and then performed an unbiased cluster analysis. This allowed, at both P0 and P7, to distinguish two main cell subpopulations, classified as BG progenitors or parenchymal astrocyte progenitors, as determined by the expression of well-known markers (Figures S1 and S2, see Methods). Furthermore, a subpopulation of cells expressing typical markers of cerebellar neurogenic progenitors, such as *Ptf1a*, *Dcx*, and *Ascl1* (Gleeson et al., 1999; Grimaldi et al., 2009; Hoshino et al., 2005), was identified at P0 (Figure S1), but disappeared at P7, as expected (Figure S2; Leto et al., 2016). Next, we compared the gene expression profiles of these three subpopulations with the gene lists of A$^-$/G$^+$ and A$^+$/G$^+$ (Figure 4b,c). At both stages, A$^-$/G$^+$ cells shared several genes with BG/BG progenitors and none with parenchymal astrocytes and their precursors (Figure 5a,b). Among these genes, beside some already highlighted above, we found growth/differentiation factor 10 (*Gdf10*) which was implicated in BG development (Mecklenburg et al., 2014) and the WNT inhibitory factor 1 (*Wif1*), recently associated with a BG subset (Kozareva et al., 2020). By contrast, A$^+$/G$^+$ cells reflected the opposite pattern (Figure 5a,b) and showed expression of angiotensinogen (*Agt*), involved in regulation of cerebral blood flow and synaptic transmission, and of the solute carrier family 6 member 11 (GAT-3) (*Slc6a11*), a modulator of inhibitory transmission. Both genes were formerly shown to be expressed by astrocytes (Batiuk et al., 2020; Matos et al., 2018; Ribak et al., 1996) but here they are reported for the first time in cerebellar parenchymal astrocytes.

Interestingly, *Atp1b2* was always found among the top 100 differentially expressed genes of the parenchymal astrocyte subpopulation. Furthermore, A$^-$/G$^+$, but not A$^+$/G$^+$, cells shared two genes with the "neurogenic progenitor" subpopulation found at P0: *Ptf1a* and *Kirrel2*, both of which encode proteins involved in the GABAergic fate specification in the cerebellum (Hoshino et al., 2005; Seto et al., 2014). Overall, these results are in line with the histological data and suggest that A$^+$/G$^+$ cells may be more committed toward a parenchymal astrocyte fate and cannot produce BG. They also suggest that A$^-$/G$^+$ cells comprise progenitors for both GABAergic interneurons and the BG lineage. Supporting these hypotheses, strikingly similar results were obtained when comparing the gene expression profiles of A$^+$/G$^+$ and A$^-$/G$^+$ cells with those of adult cerebellar astrocytes, obtained from a recently published single nuclei (sn) RNA-seq data set (Kozareva et al., 2020) (Figure S3). This comparison indeed confirmed that A$^+$/G$^+$ cells are transcriptionally similar to adult parenchymal astrocytes and share no similarities with BG, while A$^-$/G$^+$ cells display opposite features (Figure 5c).

Combining information obtained from single cell data sets with our transcriptomic data (Figures S1–S3, 4 and 5a–c), we selected candidate differentially expressed genes distinguishing different astrocyte-like progenitor clusters (Figure 5a–c) and tested their protein expression *in vivo* on P3 cerebellar sections. Using antibody staining we validated the presence of GRIA1, NPY, and MYBPC1 proteins in A$^-$/G$^+$ BG progenitors in the PCL (Figure 5d–f). Of note, we found MYBPC1 to be also enriched in A$^-$/G$^+$ cells in the PWM compared to A$^+$/G$^+$ progenitors (Figure 5 g). By contrast, AGT and SLC6A11, which were identified in the parenchymal astrocyte cluster, showed a protein expression that was confined to A$^+$/G$^+$ cells of the PWM (Figure 5h,i). In summary, our results support the notion that A$^+$/G$^+$ and A$^-$/G$^+$ cells are two transcriptionally different astroglial precursor populations with distinct differentiation potentials.

## 3.5 | Grafting experiments uncover distinct differentiation potentials between A$^+$/G$^+$ and A$^-$/G$^+$ precursors

To directly assess the actual developmental potential of A$^+$/G$^+$ and A$^-$/G$^+$ cells, we performed grafting experiments. We isolated A$^+$/G$^+$ and A$^-$/G$^+$ cells from P1–P3 β-actin-GFP cerebella, a stage of intense genesis of interneurons and glia cells (Leto et al., 2016). A$^+$/G$^+$ or A$^-$/G$^+$ cells were injected into the cerebellar vermis of

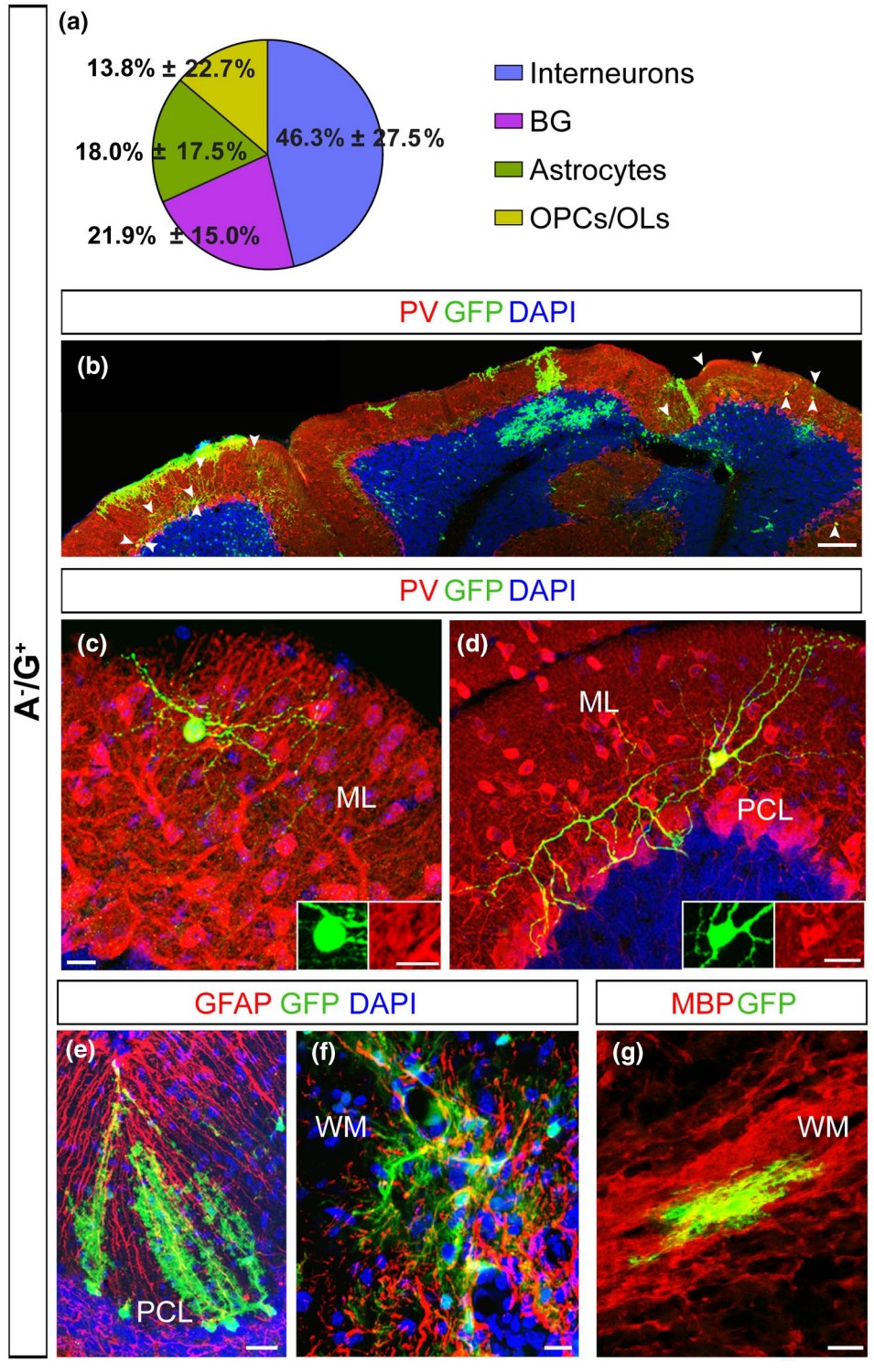

**FIGURE 6** Cell transplantation reveals a multipotent *in vivo* differentiation potential for A⁻/G⁺ precursors. (a–g) A⁻/G⁺ cells isolated from β-actin-GFP⁺ cerebella were injected into the cerebellum of β-actin-GFP⁻ mice at P1–P3. (a) A pie chart presents the average percentages of GFP⁺ interneurons, BG, astrocytes and oligodendrocytes among all GFP⁺ cells in A⁻/G⁺ hosts. (b) Low magnification images reveal the presence of interneurons in the ML of A⁻/G⁺ grafts (arrowheads in b). (c,d) High magnification pictures (as well as inserts in c and d) illustrate the co-expression of GFP and PV. (e–g) Transplanted A⁻/G⁺ cells give also rise to different glial cells: BG in the PCL (e), parenchymal astrocytes (f) and MBP⁺ OLs (g) in the WM. Scale bars: 100 µm (b); 20 µm (e–g); 10 µm (c,d). Sample group size: (b,c,e–g (*n* = 9); d (*n* = 7)). BG, Bergmann glia; GL, granular cell layer; ML, molecular layer; OPC/OL, oligodendrocyte precursor cells/oligodendrocytes; PCL, Purkinje cell layer; PV, Parvalbumin; WM, white matter [Color figure can be viewed at wileyonlinelibrary.com]

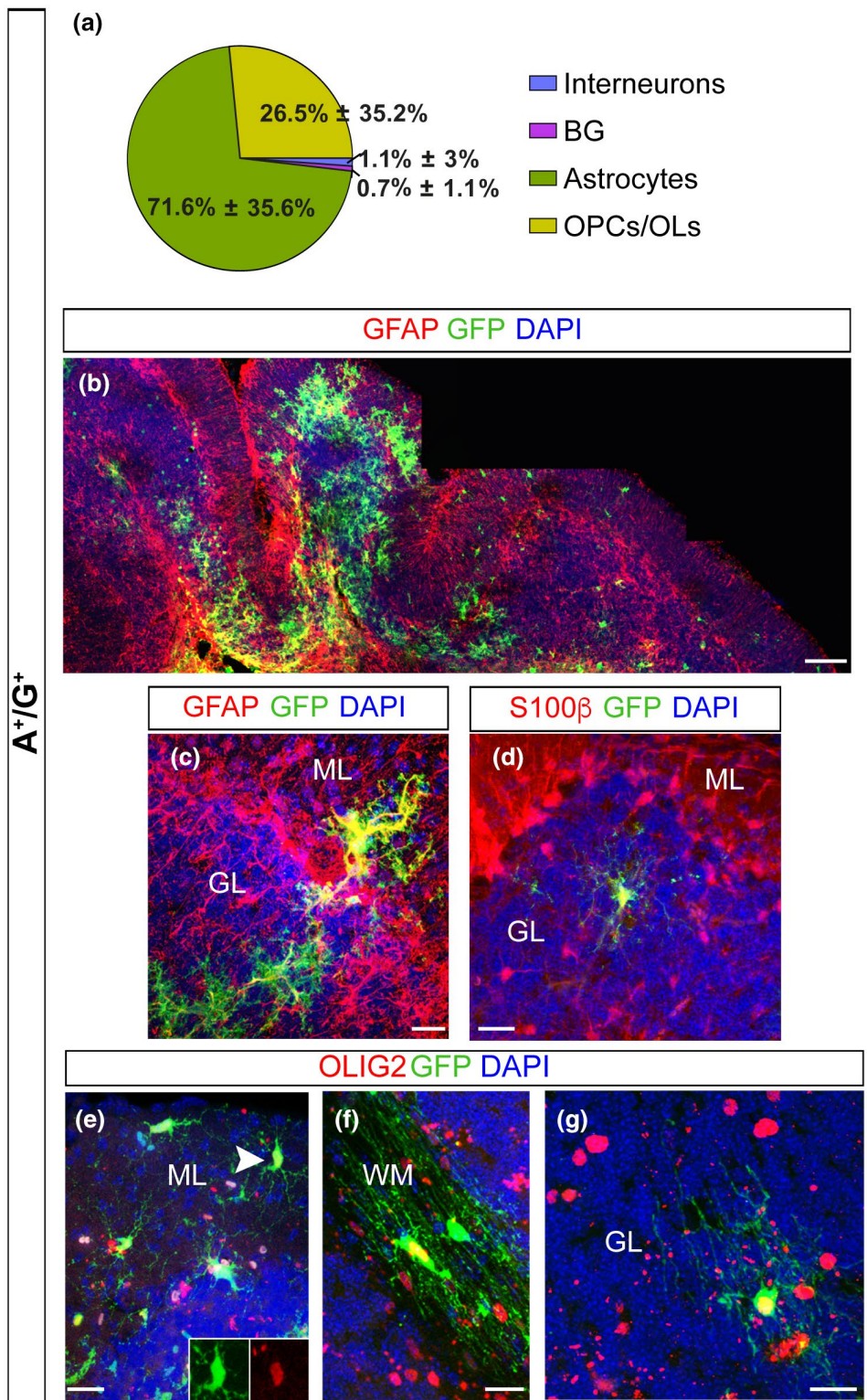

**FIGURE 7** Cell transplantation reveals glial restricted *in vivo* differentiation potential for A⁺/G⁺ precursors. (a–g) A⁺/G⁺ cells were isolated from β-actin-GFP⁺ cerebella (see Methods) and injected into the cerebellum of β-actin-GFP⁻ litter mates at P1–P3 (see Methods). (a) A pie chart presents the average percentages of GFP⁺ interneurons, BG, astrocytes and oligodendrocytes, among all GFP⁺ cells in the A⁺/G⁺ hosts. (b) Low magnification images illustrate the absence of interneurons and BG in the ML and instead the presence of glial cells throughout all the cerebellar layers. (c–g) A⁺/G⁺ transplants revealed that only parenchymal astrocytes (c,d) and OPCs/OLs (e–g), but no BG nor interneurons are generated in the host tissue after transplantation. The arrowhead and the insert in e point to a GFP⁺OLIG2⁺ OPC in the ML. Scale bars: 100 µm (b); 20 µm (c–g). Sample group size: (b (*n* = 9); c–g (*n* = 7)). BG, Bergmann glia; GL, granular cell layer; ML, molecular layer; OPC/OL, oligodendrocyte precursor cells/oligodendrocytes; PCL, Purkinje cell layer; WM, white matter [Color figure can be viewed at wileyonlinelibrary.com]

P3 β-actin-GFP⁻ littermates and tracked by the expression of GFP. For the analyses, the cellular phenotype was defined according to co-expression of glial or interneuronal markers, as well as by cell morphology and layering. The different cell populations grafted showed remarkable differences in frequencies of astrocytes and BG ($p < 0.0001$; Pearson's Chi-squared test; Table 5) as well as significantly different amounts of glia and interneurons ($p < 0.0001$; Pearson's Chi-squared test; Table 5) (Figures 6a,b and 7a,b). In both conditions, the grafted cells differentiated into glial cells, including a fraction of oligodendrocytes, as defined by OLIG2 and MBP expression and morphology ($A^+/G^+$: 26.56% ± 11.1%; $A^-/G^+$: 13.75% ± 7.1%; Figures 6g and 7e–g) and a major proportion of GFAP⁺ astrocytes (Figures 6e,f and 7c,d). Further analysis showed that astrocytes from $A^+/G^+$ and $A^-/G^+$ (tot astrocytes $A^+/G^+$: 71.61% ± 11.2% versus. $A^-/G^+$: 18.0% ± 5.4%; Figures 6b and 7b) were similarly distributed in the GL and the WM. However, and very interestingly, only astrocytes derived from $A^-/G^+$ cells were found in the ML (Figure 6b) where they differentiated into astrocytes including BG (Figure 6a,b,e) which were not seen in the $A^+/G^+$ transplants ($A^-/G^+$: 21.9% ± 4.8% vs. $A^+/G^+$ 0.7% ± 0.3%) (Figure 7a,b). A further striking difference in the progenies of the grafted cells is the yield of interneurons—these were exclusively found in the $A^-/G^+$ grafts (Figures 6a-d and 7a,b). The ML comprises two populations of PARVALBUMIN⁺ (PV) interneurons: basket cells and stellate cells. Both cell types are marked by the expression of PV and are generated at the early postnatal period, precisely at the age we chose for the grafting (Leto et al., 2006). Basket and stellate cells were found in the ML of the $A^-/G^+$ grafting sides (arrowheads in Figure 6b, stellate cells: Figure 6c, basket cells: Figure 6d). Overall, the transplantation experiment revealed a clearly distinct differentiation potential of $A^-/G^+$ and $A^+/G^+$ progenitor populations: $A^-/G^+$ cells differentiated into interneurons and all major cerebellar astrocyte phenotypes including BG while $A^+/G^+$ were exclusively gliogenic and, among astrocytes, differentiated only into GL and WM parenchymal astrocytes.

To expand further on the distinct differentiation capacities of the precursor populations we assessed their potential in a non-neurogenic environment. After heterochronic grafting of neonatal $A^-/G^+$ or $A^+/G^+$ cells into P60 cerebella, we detected PV positive interneurons exclusively in the $A^-/G^+$ grafts (Figure S6a). By contrast, $A^+/G^+$ grafted cells differentiated exclusively into glial cells (Figure S6b). These results confirm the multipotential of $A^-/G^+$ cells and the distinct developmental potential for $A^-/G^+$ and $A^+/G^+$ cells.

In summary, early (GLAST⁺) cerebellar progenitors that do not express ACSA-2 retain a remarkably broad differentiation potential, as highlighted by the production of interneurons, and by the ability to differentiate into all major cerebellar astrocyte types including BG. However, a subpopulation of GLAST⁺ precursors that co-expresses ACSA-2 defines a population of essentially glial-restricted progenitors that differentiate into defined astrocyte types.

# 4 | DISCUSSION

The heterogeneity of cerebellar astrocytes is still insufficiently characterized although more and more information, especially using bulk or single cell sequencing, is arising (Carter et al., 2018; Kozareva et al., 2020; Mizrak et al., 2019; Peng et al., 2019; Swartzlander et al., 2018; Vladoiu et al., 2019; Zeisel et al., 2018). Here, using surface marker profiling, gene expression analyses and cell , we present the subclassification of GLAST⁺ cerebellar precursors. The dual presence of ACSA-2 and GLAST was associated with a non-neurogenic phenotype giving rise to parenchymal astrocytes and oligodendrocytes whereas the absence of ACSA-2 was coupled with a multipotent potential leading to BG, parenchymal astrocyte, oligodendrocytes, and interneuron cell fates.

## 4.1 | BG precursors do not express ACSA-2

In the developing cerebellum, GLAST is detectable on radial glia cells as early as E12 and shows a broad expression on all astrocytes of the neonatal and adult cerebellum (Mori et al., 2006). In contrast to GLAST and other pan-astrocytic markers, we found ACSA-2 to be restricted in the PWM and not expressed in BG precursors of the PCL in the early neonatal cerebellum (Figure 2). In the adult cerebellum, ACSA-2 is co-expressed with GLAST on a proportion of WM astrocytes and velate astrocytes of the GL (Figure 1). Since ACSA-2 showed areas of bright and low intensities local circuits might control the protein expression. Interestingly, BG, a third type of astrocytes, which differentiates from precursors of the PCL, showed low levels of ACSA-2 and no labeling of the typical lamellate processes. Moreover, BG are only generated upon transplantation of $A^-/G^+$ precursors that share a gene expression profile previously described for BG progenitors (Kozareva et al., 2020; Vladoiu et al., 2019). For example, we identified and validated three of the top 10 differentially expressed genes: *Mybpc1*, *Npy*, and *Gria1*. As discussed in the literature, all three proteins are important for BG. We showed that these proteins are expressed on $A^-/G^+$ BG precursors (Figure 5): the first candidate, MYBPC1 was previously associated with muscle tissue and only recently connected with the BG lineage (Kozareva et al., 2020; Rodriques et al., 2019). MYBPC1⁺ cells were also detected in the PMW where it is enriched in $A^-/G^+$ cells, supporting an association between BG and multipotent precursors. Neuropeptide Y (NPY) is another protein we validated on BG precursors. NPY was demonstrated to be present on BG before (Reeber et al., 2018). It serves as a trophic factor and thereby potentially modulates adult neurogenesis (Decressac et al., 2009; Geloso et al., 2015; Lattanzi & Geloso, 2015). Finally, we validated the presence of the glutamate ionotropic receptor AMPA type subunit 1 (*Gria1*) on $A^-/G^+$ cells. GRIA1, also known as GluA1, is one of the predominant AMPA receptors expressed by BG and was shown to be perquisite for proper motor coordination (Saab et al., 2012). Nephrin is also

known as being expressed by BG precursors in the PCL (Putaala et al., 2001). In addition, GDF10, involved in BG development (Gupta et al., 2018; Mecklenburg et al., 2014) and GLI1, expressed by BG during development and adulthood suggesting activation of Shh signaling (Corrales et al., 2004; Fleming et al., 2013), were further upregulated in the $A^-/G^+$ population. Interestingly, it has been shown that the specification of BG is driven by extrinsic signals that lead to intrinsic changes (De Luca et al., 2016; He et al., 2018; Leto et al., 2016). Our data therefore support this idea, suggesting that several genes specific for the BG lineage might be suppressed in $A^+/G^+$ precursors, and further stresses the fact that BG precursors are unlikely to be $A^+/G^+$. However, a full demonstration would need further investigation.

## 4.2 | Cerebellar $A^-/G^+$ precursors are multipotent while $A^+/G^+$ precursors are glia restricted

The PWM of the developing cerebellum is composed of distinct progenitor subsets namely multipotent cells and fate-restricted precursors (Buffo & Rossi, 2013; Leto et al., 2010; Milosevic & Goldman, 2002). Several studies have used lineage tracing or marker gradients to address the heterogeneity of these precursor populations (Cerrato et al., 2018; Fleming et al., 2013; Parmigiani et al., 2015). Remarkable, not all $GLAST^+$ cells in the PWM are $ACSA-2^+$, thus arguing that ACSA-2 marks a subpopulation of $GLAST^+$ cells. Comparable to previous studies describing $GLAST^+$ precursors as bipotent and thus capable of developing glia cells and interneurons (Parmigiani et al., 2015), we show that $A^-/G^+$ precursors differentiate into interneurons and all astrocyte types including BG. Accordingly, only $A^-/G^+$, but not $A^+/G^+$, precursors, express genes, such as *Ptf1a* and *Kirrel2*, found specifically in a cluster of neurogenic astroglial progenitors that is transcriptionally different from BG and parenchymal astrocytes (Kozareva et al., 2020; Vladoiu et al., 2019). By contrast, progenitors that co-express GLAST along with ACSA-2 ($A^+/G^+$) showed a limited differentiation potential *in vivo*. With the notable exception of BG, $A^+/G^+$ cells generated exclusively glia and did not give rise to interneurons. These findings highlight that $A^+/G^+$ progenitors have a purely gliogenic character. Indeed, $A^+/G^+$ progenitors display a gene signature of parenchymal astrocytes while they do not express any gene of neurogenic or BG progenitor clusters. We identified *Slc6a11* and *Agt* as representative genes and disclosed their protein expression pattern. Interestingly, SLC6A11 recapitulated the expression pattern of ACSA-2 to a very high degree being expressed by $A^+/G^+$ PWM precursors at P3. The expression of the *Agt* gene has been previously described to discriminate astrocytes in non-telencephalic and caudal regions (Zeisel et al., 2018). We could confirm and further expand this finding by showing that the AGT protein is enriched in $A^+/G^+$ precursors in the PWM. Thus, our data reveal that ACSA-2 expression defines two transcriptionally and functionally distinct progenitor populations in the developing cerebellum.

## 4.3 | The relationship between $A^+/G^+$ and $A^-/G^+$ precursors

The $A^-/G^+$ population might serve as a progenitor pool giving rise to a common precursor lineage, which then develops interneurons and all astroglial subtypes. However, there is currently no clonal study that proves the existence of such lineage in the postnatal cerebellum. As a summary of this and previous studies (Cerrato et al., 2018; Parmigiani et al., 2015) we propose that $GLAST^+$ precursors constitute a heterogeneous precursor pool.

The first progenitor population ($A^-/G^+$) is multipotent and less committed. This population resides in the PWM and includes progenitors generating both interneurons, BG and WM astrocytes (Parmigiani et al., 2015). $A^+/G^+$ precursors instead contain astrocyte progenitors already committed to becoming astrocytes that generate WM/GL cells. Interestingly, however, both $A^+/G^+$ and $A^-/G^+$ cells give rise to oligodendrocytes.

Earlier fate-mapping studies showed that oligodendrocytes are generated from $GLAST^+$ progenitors preferentially in the embryo (Parmigiani et al., 2015). However, we obtained oligodendrocytes in our grafts upon the transplantation of early postnatal $GLAST^+$ cells.

Apart from the frequencies of oligodendrocytes, the proportional contribution of BG, interneurons, and astrocytes in the $A^-/G^+$ grafts is comparable with the frequencies seen in Parmigiani et al. (2015). Since $ACSA-2^+$ and $GLAST^+$ cells did not co-express PDGFRα or NG2, we consider the possibility of contaminating oligodendrocyte precursors within the grafts as rather unlikely. In this study we used an antibody affinity-based approach, whereas the former study used an inducible GLAST::Cre$^{ERT}$ line and lentiviral constructs to label $GLAST^+$ precursors. Thus, the variances might be explainable by the different methods used, as the promoter activity might not reflect the intrinsic level of the protein.

Despite their dissimilarity we showed in this study that both populations—$A^+/G^+$ and $A^-/G^+$—express GLAST, thereby suggesting a lineage relationship. It might be that astroglial-like $A^-/G^+$ cells of the PWM give rise to astroglial-committed $A^+/G^+$ progenitors. Future experiments will disclose how these subsets are interrelated and whether $A^+/G^+$ cells are ontogenically related to $A^-/G^+$ cells or form a separate class.

## 4.4 | Outlook

Na$^+$/K$^+$ ATPases maintain sodium and potassium concentrations, balance osmosis, and preserve the electrochemical gradient (Lecuona et al., 1996). The target of the anti-ACSA-2 antibody, the beta-2 subunit of the sodium/potassium ATPase (ATP1B2), is highly enriched in the murine cerebellum and is stably expressed during stab wound injury (Batiuk et al., 2017). This makes ACSA-2 an ideal candidate to not only study its function during cerebellar development but also the biological role upon neuro-inflammatory diseases and degenerative disorders.

Recently, a structural variation in exon 2 of the *ATP1B2* gene has been linked to cerebellar ataxia in Belgian Shepherds (Mauri

et al., 2017). Their publication described the same defects in motor coordination as seen in ATP1B2$^{(0/0)}$ mice accompanied with death in the second postnatal week (Antonicek & Schachner, 1988; Magyar et al., 1994). Furthermore, aberrant expression of ATP1B2 has been linked to glioblastoma multiforme (GBM). It was shown that patients with GBM have changes in the isoforms of ATP1B2 (Rotoli et al., 2017) and that targeting ATP1B2 induced glioblastoma cells apoptosis (Li et al., 2019). These examples emphasize the importance to further investigate the function of ACSA-2/ATP1B2 expressing astrocytes that was beyond the scope of this study.

## DECLARATION OF TRANSPARENCY

The authors, reviewers and editors affirm that in accordance to the policies set by the *Journal of Neuroscience Research*, this manuscript presents an accurate and transparent account of the study being reported and that all critical details describing the methods and results are present.

## ACKNOWLEDGMENTS

This study was supported by local funds of the University of Turin to AB and by Ministero dell'Istruzione, dell'Università e della Ricerca—MIUR project "Dipartimenti di Eccellenza 2018–2022" to Dept. of Neuroscience "Rita Levi Montalcini." VC was supported by the Umberto Veronesi Foundation and the IBRO-PERC inEurope Short Stay fellowship. In addition, we acknowledge our animal caretaker, Paul Colesar, for his support with the breeding of the mice used in this study.

## CONFLICT OF INTEREST

S.T., M.K., M.J., and A.Bo. are employees of Miltenyi Biotec B.V. & Co. KG. The remaining authors declare no competing financial interests.

## AUTHOR CONTRIBUTIONS

Conceptualization: C.G.K., A.Bo., E.P., V.C. and A.B. Methodology: C.G.K., E.P., V.C. Validation: C.G.K, E.P. Investigation: C.G.K., E.P., V.C. Formal Analysis: C.G.K., E.P., V.C., S.T. Software: V.C., S.T., M.K.; Data Curation: V.C., S.T. Visualization: C.G.K., E.P., V.C., S.T. Writing – Original Draft Preparation: C.G.K., E.P., M.J., S.T., A.B., A.Bo. Writing – Review & Editing: C.G.K., E.P., V.C., S.T., M.J., A.B., A.Bo. Resources: M.J., A.B., A.Bo. Supervision: M.J., A.B., A.Bo. Funding Acquisition: A.B.

## PEER REVIEW

The peer review history for this article is available at https://publons.com/publon/10.1002/jnr.24842.

## DATA AVAILABILITY STATEMENT

The gene expression profiling data that support the findings of this study are available in NCBI's Gene Expression Omnibus (RRID:SCR_005012) and are accessible through GEO Series accession number GSE117886. Further data that support the findings of this study are available from the corresponding author upon reasonable request.

## ORCID

*Christina Geraldine Kantzer* (iD) https://orcid.org/0000-0002-4975-7579
*Elena Parmigiani* (iD) https://orcid.org/0000-0002-3433-1823
*Melanie Jungblut* (iD) https://orcid.org/0000-0001-9126-6610

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

## SUPPORTING INFORMATION

Additional Supporting Information may be found online in the Supporting Information section.

**FIGURE S1** Identification of distinct astrocyte subpopulations in scRNA-seq data sets of P0 cerebella. Seurat clustering of the astrocyte-like cells in a data set of P0 cerebella resulted in clearly segregated subpopulations that could be classified according to the expression of known marker genes (see Methods). The expression of BG-enriched genes such as *Gdf10*, *Gria1*, and *Gria4* allowed to identify the BG/BG progenitor subpopulations (cluster n. 0). The expression of *Aqp4* was used to identify the parenchymal astrocytes/astrocyte progenitor population (cluster n. 2). Moreover, the comparison of the gene expression signatures of each of these two populations

across ages (see also Figures S2 and S3) confirmed the classification as a good fit. At P0, cluster n.1 expressed genes typically associated with a neuronal progenitor fate, such as *Ptf1a*, *Dcx*, and *Ascl1* (Hoshino et al., 2005; Gleeson et al., 1999, Grimaldi et al., 2009), and was therefore classified as a cluster of neurogenic progenitors. Further clusters did not present a clear identity—based on known marker genes—and were therefore not further considered for the analyses

**FIGURE S2** Identification of distinct astrocyte subpopulations in scRNA-seq data sets of P7 cerebella. Seurat clustering of the astrocyte-like cells in a data set of P7 cerebella resulted in clearly segregated subpopulations that could be classified according to the expression of known marker genes (see Methods). The expression of BG-enriched genes such as *Gdf10*, *Gria1*, and *Gria4* allowed to identify the BG/BG progenitor subpopulations (cluster n.1). The expression of *Aqp4* was used to identify the parenchymal astrocytes/astrocyte progenitor population (cluster n.0). The neurogenic progenitor cluster identified at P0 (Figure S1) was not detected at P7, and correlates with the expected downregulation of neurogenic genes (*Ptf1a*, *Dcx*, and *Ascl1*) at this age. Further clusters did not present a clear identity—based on known marker genes—and were therefore not considered for the analyses

**FIGURE S3** Identification of distinct astrocyte subpopulations in snRNA-seq data sets of adult cerebella. Seurat clustering of the astrocyte-like cells in data set of a P60 ("Adult") cerebella resulted in clearly segregated subpopulations that could be classified according to the expression of known marker genes (see Methods). The expression of BG-enriched genes such as *Gdf10*, *Gria1*, and *Gria4* allowed to identify the BG/BG progenitor subpopulations (cluster n. 0). The expression of *Aqp4* was used to identify the parenchymal astrocytes/astrocyte progenitor population (cluster n. 1). The neurogenic progenitor cluster identified at P0 (Figure S1) was not detected in the adult cerebellum

**FIGURE S4** ACSA-2 is not expressed by neonatal cerebellar oligodendrocytes but is expressed by a subpopulation of astroglial-like progenitors. (a) Confocal stacks of P3 cerebellum confirmed ACSA-2 expression to be confined to cerebellar astrocytes in the PWM ACSA-2 is not co-expressed with the oligodendrocyte marker NG2 (a–a′″; filled arrowheads point to NG2$^+$/ACSA-2$^-$ cells). (b) Dissociated cerebellar samples were analyzed by flow cytometry for the co-expression of ACSA-2 and the oligodendrocyte precursor cell marker platelet-derived growth factor receptor alpha (PDGFRα). The overlay was less than 2% for both populations at P1 ((A$^+$/G$^+$): 1.68%; (A$^-$/G$^+$): 1.22%) and at P3 ((A$^+$/G$^+$): 0.4517%; (A$^-$/G$^+$): 0.52%)). Frequencies of A$^-$/G$^+$ and A$^+$/G$^+$ cells described in Figure 2a–c were quantified over total cells (c) or total BLBP$^+$ astroglial-like progenitors (d) in the different layers of a P3 cerebella (i.e., PWM, GL, PCL/ML). Frequencies are presented also as pie charts for each cerebellar layer. At this stage A$^+$/G$^+$ cells are exclusively present in the PWM where they represent about half of the astroglial-like progenitors. Scale bar: 100 μm. Nuclear stain: DAPI. GL, granular cell layer; ML, molecular layer; PCL, Purkinje cell layer; PWM, prospective white matter

**FIGURE S5** A minor fraction of ACSA-2$^+$ precursors in the PWM proliferates. (a,b) Among the proliferating cells of the PWM a minor proportion of ACSA-2$^+$ cells ((d) 8.4% ± 1.6% ($n = 3$)) showed Ki67 positivity, thus proliferates (a and close-up in b). Orthogonal projection shows the surface marker ACSA-2 aligned around the Ki67$^+$ nuclei (c). Scale bars: 30 μm (a,b). Nuclear stain: DAPI. EGL, external granular layer; GL, granular cell layer; ML, molecular layer; PWM, prospective white matter

**FIGURE S6** A$^-$/G$^+$ precursors present a multipotent differentiation potential when transplanted into a non-neurogenic environment. (a,b) A$^+$/G$^+$ and A$^-$/G$^+$ cells were isolated from β-actin-GFP$^+$ cerebella and injected into the cerebellum of adult β-actin-GFP$^-$ mice (P60). (a) As seen by the co-labeling of GFP (a,a″) and PV (a,a′) A$^-$/G$^+$ cells generate PV$^+$ interneurons when transplanted into the adult cerebellum. (b) By contrast, A$^+$/G$^+$ cells differentiate exclusively into GFAP$^+$ astrocytes as identified by the co-staining of GFP (b,b″) with GFAP (b,b′). Scale bars: 10 μm (a,b) Sample group size: (a,b ($n = 3$)). GL, granular cell layer; ML, molecular layer; PV, Parvalbumin; WM, white matter

Transparent Peer Review Report

Transparent Science Questionnaire for Authors

