## [Transparent Peer Review Report · Journal of Neuroscience Research]

**ACSA-2 and GLAST classify subpopulations of multipotent and glial restricted cerebellar precursors
Jungblut, Melanie; Kantzer, Christina; Parmigiani, Elena; Cerrato, Valentina; Tomiuk, Stefan; Knauel,
Michail; Buffo, Annalisa; Bosio, Andreas**

Review timeline:

Submission date: 10 March 2020
 Editorial Decision: Major Modification (2 May 2020)
 Revision Received: 8 January 2021
 Editorial Decision: Accept with Minor Edits (25 February 2021)
 Revision Received: 12 March 2021
 Accepted: 17 March 2021

Editor 1: Stephen Crocker
 Editor 2: Junie Warrington
 Editor 3: David McArthur
 Reviewer 1: Cory Willis
 Reviewer 2: Richard Milner

1st Editorial Decision

Decision letter
 Jungblut:

Dear Dr

Thank you for submitting your manuscript to the Journal of Neuroscience Research. We have now received the reviewer feedback and have appended those reviews below. As you will see, the reviewers find the question addressed to be of potential interest. Yet, they do not find the manuscript suitable for publication in its current form.

If you feel that you can adequately address the concerns of the reviewers, you may revise and resubmit your paper within 60 days. It will require further review. Please explain in your cover letter how you have changed the present version. If you require longer than 60 days to make the revisions, please contact Dr Junie Warrington (jpwarrington@umc.edu). You can submit your revised manuscript directly by clicking on the following link: *** PLEASE NOTE: This is a two-step process. After clicking on the link, you will be directed to a webpage to confirm. ***

https://mc.manuscriptcentral.com/jnr?URL_MASK=d41b3844a14c415bb4f827bbabb2e65a

Thank you again for your submission to the Journal of Neuroscience Research; we look forward to reading your revised manuscript.

Best Wishes,

Dr Stephen Crocker
 Associate Editor, Journal of Neuroscience Research

Dr Junie Warrington
 Editor-in-Chief, Journal of Neuroscience Research

Editorial Comments to Author:

1. Please incorporate Research Resource Identifiers (RRIDs) in your citation of all resources used in your manuscript (antibodies, software tools, databases, model organisms) where applicable in the text, exactly as you would a regular citation or Genbank Accession number. Please also be sure these RRIDs are included in your keywords list in addition to the required keywords. For any antibodies, we ask that you

also include the RRIIDs in your antibody table, in addition to citing them in the text. An example of how to list RRIIDs in your antibody table can be found in the example antibody table attached and for more information about how to obtain and cite RRIIDs within your text, please visit the "Resource Identification Initiative" section of our author guidelines: [http://onlinelibrary.wiley.com/journal/10.1002/\(ISSN\)1097-4547/homepage/ForAuthors.html](http://onlinelibrary.wiley.com/journal/10.1002/(ISSN)1097-4547/homepage/ForAuthors.html)

Furthermore, in accord with JNR requirements, you will have to provide additional information on each of the antibodies that you use. This must include the source (not just the manufacturer and catalog or lot number, but the species it was raised in, and the EXACT structure of the immunizing antigen - including the amino acid sequence for peptide antigens); characterization (e.g., what does it recognize on immunoblots); and appropriate controls (i.e. the effect of blocking peptides for immunohistochemical localization).

2. To enable readers to locate archived data from Journal of Neuroscience Research papers, we require authors to include a 'Data Accessibility' section just before the References. This should list the database(s) and URL(s) or dataset DOIs for all data associated with the manuscript. Data deposit repositories might include unstructured repositories such as Dryad, FigShare, NeuroMorpho or centralized repositories from the institutions in which the research was conducted. We also strongly recommend depositing data in the Open Science Framework. JNR will also allow small data sets to be included as Supplementary Files with the article.

3. JNR does not allow the use of bar and line graphs for continuous data, as many different data distributions can lead to the same bar or line graph, and representation of the full data may suggest different conclusions (Weissgerber et al., 2017). The following are the new guidelines that authors should take into consideration when graphically illustrating data: For continuous data, JNR requires authors to use univariate scatterplots, violin plots or boxplots. If boxplots are used, their component parts need to be clearly identified in the caption. Use dotplot overlays if sample size is small.

Associate Editor: Crocker, Stephen

Comments to the Author:

Both reviewers have expressed enthusiasm for the study and indicate that the findings reported are highly relevant. Specific suggestions were made by both reviewers and the authors should address these comments with additional analyses, and experimentation, where necessary. In particular, the validation of select genes identified in the bulk RNAseq analysis would be considered necessary to verify the findings reported in this study.

Reviewer: 1

Comments to the Author

The manuscript by Kantzer and Parmigiani et al. sets out to phenotype cerebellar progenitors possessing astroglia-like characteristics in the early post-natal brain to better understand their developmental fates. The authors first confirmed previous findings of ASCA-2 and GLAST expression on astroglial populations in the adult cerebellum. Building from these findings, the authors performed a more detailed and high resolution analyses of these astroglial populations in the developing cerebellum, which led to the identification of two distinct astroglia precursor populations: ASCA-2+GLAST+ and ASCA-2-GLAST+. Further profiling revealed these two populations can be separated based on both surface marker expression using FACS and transcriptomic analysis. Interestingly, over the postnatal development period studied (P1-P7) the authors observed significant decreases in the frequency of each precursor population; seemingly tied to their differentiation into more mature cell types. To identify the developmental fates of these precursors, the authors isolated cells from each population (ASCA-2+GLAST+ and ASCA-2-GLAST+) during a time of intense neuro and gliogenesis in postnatal pups (P1-P3) to graft into the developing cerebellum of littermates. This uncovered distinct differentiation potentials of the two populations, which was supported by the transcriptomic analysis. ASCA-2+GLAST+ progenitor cells differentiated into parenchymal astrocytes whereas the ASCA-2-GLAST+ progenitor pool had a more diverse developmental fate differentiating into parenchymal astrocytes, Bergmann glia, and interneurons. Surprisingly, these precursor populations maintained their distinct differentiation capacities following engraftment into the adult brain, suggesting the maintenance of this progenitor pool in the cerebellum into and throughout adulthood.

The overall study is well-designed and provides a necessary foundation which will allow for more in-depth and thorough future examinations of these unique cell-populations during development, in adulthood, and following neurological disorders. This manuscript is a necessary, critical, and timely study that provides a

useful resource for future investigators to utilize as a guide in their own work.

However, issues remain in the manuscript that must be addressed:

1. It is necessary and required the authors include quantification of the unique expression patterns of their markers within the anatomical locations profiled in figure 2A,B,C relative to the total population of cells. This is important to discern how abundant this population is relative to other cell types (or progenitor/precursor pools) in the developing cerebellum.
2. Cell transplant data from figure 5O,P indicates a permissive environment for these two progenitor populations to differentiate into their requisite cell types in the adult cerebellum. It is important and necessary to determine if these progenitor populations exist in the naïve adult cerebellum or are restricted to critical post-natal development periods. Therefore, the authors are asked to identify and quantify the existence of these two progenitor populations within the adult cerebellum.
3. The authors performed bulk RNA sequencing to identify transcriptomic differences between each of the progenitor populations. However, the data as shown presents an incomplete picture. The authors only provide a gene signature for the ASCA-2-GLAST+ progenitor cells based on the differential expression compared to the ASCA-2+GLAST+ pool. The authors must expand this figure to include the same analysis for the ASCA-2+GLAST+ progenitor cells when compared to the ASCA-2-GLAST+ pool.
4. The authors include a list of differentially expressed genes within the ASCA-2-GLAST+progenitor pool. However, it is not shown if these differentially expressed genes (or their proteins) are expressed within the progenitor pool in vivo. It is necessary the authors provide proof of the restricted expression of at least a few of these genes (or gene products) in their ASCA-2-GLAST+ progenitor pool through a complementary approach (such as FISH or IHC/IF). A similar analysis must be performed on the ASCA-2+GLAST+ progenitor pool in accordance with point 3 (above).

Minor points:

1. In the methods section on page 10, line 237 beginning with "Only probes with...", the authors state the cut-off was "...at least three-fold median up- or downregulation..." yet in the results section page 15, line 355 the authors state the cut-off was "...at least four-fold median up- or down-regulation...:". Please correct this inconsistency.
2. The authors are asked to provide a gating strategy as a supplemental figure for the flow cytometry analysis performed. This will help future investigators when determining their gating strategy.
3. In the methods section on page 8, line 183 under the heading 'Homochronic and heterochronic transplantations in vivo' the authors must include the number of cells transplanted per μ l and how the cells were delivered (i.e., single injection, two injections). This is critical information for replication of this aspect of the study in the future.
4. Throughout the manuscript the authors reference data but fail to include it either in the main figures or supplemental. The authors must provide all data referenced.
5. Minor spelling and grammatical mistakes need to be cleaned up. For example, on page 13 line 303 there is a misspelling of under as 'und'.

Reviewer: 2

Comments to the Author

This study by Kantzer et al takes a multi-modal approach to define the different types of glial progenitor in the developing cerebellum. Interestingly, in the developing cerebellum ACSA-2 is expressed by most glial progenitors and astrocytes but not by Bergmann glia precursors. Dual staining reveals that at different developmental stages, ACSA-2 and GLAST show unique expression patterns with some degree of overlap. Flow cytometry of isolated single cell populations from dissociated cerebellar tissue identify two main populations of progenitors: G+/A+ and G+/A-. Gene expression profiling show that these two separate populations expression differential patterns of gene expression. Most strikingly, transplantation into developing cerebella demonstrated that the two populations of progenitors have markedly different differentiation potential, with G+/A+ cells giving rise to only glial cells, namely oligodendrocytes and

astrocytes, while in contrast, G+/A- cells gave rise to glial cells as well as a high percentage of neurons. Overall, I think this is an important and well-executed study that provides important new information to the field of cerebellar development. Some minor questions remain:

1. In both the abstract and significance statement, the phrase that says "neurogenic or glial-restricted" is rather misleading because it implies that one population is neurogenic-restricted, which isn't actually the case. Surely it would be better to describe these population as still multipotential?
2. In Figure 1 legend, it describes panel D but there is no panel D there.
3. In the flow cytometry studies, information should be provided as to the survival of cells through the isolation and labeling procedure. Were dead cells excluded? This is important because there is always the possibility that certain cell populations are more hardy and more likely to survive the isolation and labeling procedures and thus bias the results one way.
4. In the grafting studies, while we are told that a high % of cells survived the grafting process, no information is provided what this actually means. Was this 90% or 9%? This should be corrected.
5. At the same time, it was not immediately clear what age the donor mice were receiving the grafts. This should be spelled out to make it clear.
6. Please correct page 12, line 313, change "extend" to "extent."
7. Same page, line 318, change "on" to "at."

Authors' Response

Dear Dr. Warrington, dear Dr. Crocker, dear reviewers,
we would like to thank the reviewers for their valuable comments and the editor for the extension of the re-submission period.

As follows we will address the comments of the editor and reviewers point by point.

Comments of the editor (in italics):

1. Please incorporate Research Resource Identifiers (RRIDs) in your citation of all resources used in your manuscript (antibodies, software tools, databases, model organisms) where applicable in the text, exactly as you would a regular citation or Genbank Accession number. Please also be sure these RRIDs are included in your keywords list in addition to the required keywords. For any antibodies, we ask that you also include the RRIDs in your antibody table, in addition to citing them in the text. An example of how to list RRIDs in your antibody table can found in the example antibody table attached and for more information about how to obtain and cite RRIDs within your text, please visit the "Resource Identification Initiative" section of our author guidelines:

[http://onlinelibrary.wiley.com/journal/10.1002/\(ISSN\)1097-4547/homepage/ForAuthors.html](http://onlinelibrary.wiley.com/journal/10.1002/(ISSN)1097-4547/homepage/ForAuthors.html)

Furthermore, in accord with JNR requirements, you will have to provide additional information on each of the antibodies that you use. This must include the source (not just the manufacturer and catalog or lot number, but the species it was raised in, and the EXACT structure of the immunizing antigen - including the amino acid sequence for peptide antigen); characterization (e.g., what does it recognize on immunoblots); and appropriate controls (i.e. the effect of blocking peptides for immunohistochemical localization).

→ We have now included RRID in the keywords. In addition, all RRIDs can now be found in the manuscript text as well as in Table 1.

2. To enable readers to locate archived data from Journal of Neuroscience Research papers, we require authors to include a 'Data Accessibility' section just before the References. This should list the database(s) and URL(s) or dataset DOIs for all data associated with the manuscript. Data deposit repositories might include unstructured repositories such as Dryad, FigShare, NeuroMorpho or centralized repositories from the institutions in which the research was conducted. We also strongly recommend depositing data in the Open Science Framework. JNR will also allow small data sets to be included as Supplementary Files with the article.

→ We have now added the 'Data Accessibility' section.

3. JNR does not allow the use of bar and line graphs for continuous data, as many different data distributions can lead to the same bar or line graph, and representation of the full data may suggest different conclusions (Weissgerber et al., 2017). The following are the new guidelines that authors should take into consideration when graphically illustrating data: For continuous data, JNR requires authors to use univariate scatterplots, violin plots or boxplots. If boxplots are used, their component parts need to be clearly identified in the caption. Use dotplot overlays if sample size is small.

→ We have changed the graphs accordingly.

Comments of Reviewer 1 (in italics):

1. It is necessary and required the authors include quantification of the unique expression patterns

of their markers within the anatomical locations profiled in figure 2A,B,C relative to the total population of cells. This is important to discern how abundant this population is relative to other cell types (or progenitor/precursor pools) in the developing cerebellum.

→ We understand the point raised by the reviewer and agree that this is a valuable piece of information, that was missing. We have now quantified the proportion of the A+/G+ and A-/G+ cell populations relative to all DAPI+ cells and to the astroglial precursors to which they belong (using a co-staining with BLBP, previously shown to label astroglial-like progenitors (Cerrato et al., 2018)). As suggested by the reviewer, we have also quantified their distribution in the anatomical locations described in figure 2A, B, C, i.e. PWM, GL and PCL/ML.

Revised part of the manuscript: Figure S2

2. Cell transplant data from figure 5 O,P indicates a permissive environment for these two progenitor populations to differentiate into their requisite cell types in the adult cerebellum. It is important and necessary to determine if these progenitor populations exist in the naïve adult cerebellum or are restricted to critical post-natal development periods. Therefore, the authors are asked to identify and quantify the existence of these two progenitor populations within the adult cerebellum.

We thank the reviewer for raising such an interesting point. As described in figure 1 and figure 2 E, F, later during development the ASCA-2-negative population becomes restricted to the PCL where mature BG reside, while in the rest of the cerebellum all the GLAST+ astroglial cells differentiate into mature astrocytes and express ASCA-2. According to multiple studies, cerebellar neurogenic and astroglial progenitors are restricted to the first postnatal week, after which neurogenesis ceases and only a limited number of astrocytes are generated until the end of cerebellar development (Silbereis et al., 2009; Leto et al., 2016; Parmigiani et al., 2015; Cerrato et al., 2018). The analysis of two different recent sc/snRNAseq datasets we have now added to the manuscript also confirms that the neurogenic progenitors cluster identified in the postnatal brain disappeared in the adult cerebellum. At this age, only parenchymal and BG astrocytes are present and they express markers of mature astrocytes. Although cerebellar astrocytes seem to maintain a certain degree of plasticity in some pathological conditions (Lafarga et al., 1998; Milenkovic et al., 2005; Yao et al., 2020; Cerrato 2020), to our knowledge no data are available demonstrating the existence of active progenitor populations in the adult cerebellum in a physiological context (Leto et al., 2016). Only a rare subpopulation of Sox2+S100β⁻ cells in the PCL has been recently proposed to give rise to NeuN+ granule cells (Ahlfeld et al., 2017). However, while the location of these cells would correspond to ASCA-2-negative BG, their lack of BLBP and S100β expression suggests that they are not comprised in the GLAST+ cell subsets that we analyzed histologically or in RNAseq data.

3. The authors performed bulk RNA sequencing to identify transcriptomic differences between each of the progenitor populations. However, the data as shown presents an incomplete picture. The authors only provide a gene signature for the ASCA-2-GLAST+ progenitor cells based on the differential expression compared to the ASCA-2+GLAST+ pool. The authors must expand this figure to include the same analysis for the ASCA-2+GLAST+ progenitor cells when compared to the ASCA-2-GLAST+ pool.

→ We apologize to the reviewer. Our intention was to focus the manuscript on the ASCA-2-/GLAST+ progenitor cells that possesses a neurogenic transcriptional and functional profile. However, we understand the argument brought up by the reviewer and have now added the complementary analysis for the ASCA-2+/GLAST+ progenitor cells.

Revised part in the manuscript showing this data: Figure 4.

4. The authors include a list of differentially expressed genes within the ASCA-2-GLAST+progenitor pool. However, it is not shown if these differentially expressed genes (or their proteins) are expressed within the progenitor pool in vivo. It is necessary the authors provide proof of the restricted expression of at least a few of these genes (or gene products) in their ASCA-2-GLAST+ progenitor pool through a complementary approach (such as FISH or IHC/IF). A similar analysis must be performed on the ASCA-2+GLAST+ progenitor pool in accordance with point 3 (above).

→ We agree with the reviewer that this information is valuable to confirm the expression signatures identified. Therefore, with the aim to better define the astroglial identity of ASCA-2-/GLAST+ and ASCA-2+/GLAST+ progenitors, we interrogated published cerebellar single cell/single nucleus RNA sequencing datasets obtained early after birth and in the adult. At first in these data we identified distinct astroglial precursors/mature astrocyte

clusters, whose transcriptome was then compared to the identified target lists (Figure 4). This analysis identified 2 main astroglial progenitor clusters that correspond to different astrocytes populations in the adult cerebellum based on their gene expression profiles: parenchymal astrocytes and BG. A third astroglial-like neurogenic progenitor cluster was also identified early during development but it disappeared at P7, supporting previous *in vivo* lineage analysis studies (Silbereis et al., 2009; Fleming et al., 2013; Parmigiani et al., 2015). Comparing the expression profiles of these progenitor pools with those described in our study we found markers of parenchymal astrocytes progenitors exclusively in the ASCA-2+/GLAST+ pool and not in ASCA-2-/GLAST+ progenitor population that instead shares genes only with neurogenic and BG progenitor clusters. Based on such comparison we have also selected the most relevant differentially expressed markers (Gria1, Npy, Mybpc1 for ASCA-2-/GLAST+ progenitors and Agt, Slc6a11 for ASCA-2+/GLAST+ progenitors) and have subsequently validated them using a complementary approach (IF), as requested by the reviewer.

The modified data is now included as Figure S1 and Figure 5.

→ Dr Valentina Cerrato (University of Turin and University of Lausanne) performed the analysis of sc/sn RNAseq datasets and compared transcriptional profiles. For these reasons she is now included among the authors of this manuscript.

Minor points:

1. *In the methods section on page 10, line 237 beginning with "Only probes with...", the authors state the cut-off was "...at least three-fold median up- or downregulation..." yet in the results section page 15, line 355 the authors state the cut-off was "...at least four-fold median up- or down-regulation...". Please correct this inconsistency.*

→ We apologize for this inconsistency and have corrected the mistake.

2. *The authors are asked to provide a gating strategy as a supplemental figure for the flow cytometry analysis performed. This will help future investigators when determining their gating strategy.*

→ We have added the information that describes the gating strategy in the method section: "For the gating, we used side-scatter vs. forward-scatter to determine neural cells. Cell debris and dead cells were identified by Propidium Iodide (Miltenyi Biotec) and excluded from the analyses. After excluding doublets, the frequencies of stained cells were identified using the channel appropriate for the selected fluorophore (compare Figure 3)"

3. *In the methods section on page 8, line 183 under the heading 'Homochronic and heterochronic transplantations in vivo' the authors must include the number of cells transplanted per μ l and how the cells were delivered (i.e., single injection, two injections). This is critical information for replication of this aspect of the study in the future.*

→ We have added the information as requested by the reviewer.
Text from manuscript: "60'000 cells/ μ l, two injections of 1 μ l each"

4. *Throughout the manuscript the authors reference data but fail to include it either in the main figures or supplemental. The authors must provide all data referenced.*

→ We apologize to the reviewer and have deleted all 'data not shown' in the text.

5. *Minor spelling and grammatical mistakes need to be cleaned up. For example, on page 13 line 303 there is a misspelling of under as 'und'.*

→ We have revised the text accordingly.

Comments of Reviewer 2 (in italics):

1. *In both the abstract and significance statement, the phrase that says "neurogenic or glialrestricted" is rather misleading because it implies that one population is neurogenic-restricted, which isn't actually the case. Surely it would be better to describe these population as still multipotential?*

→ We thank the reviewer for the suggestion to describe the population as multipotent. We have changed the terminology in our manuscript accordingly.
Exemplified text from the manuscript: "These results confirm the multipotential of A-/G+ cells and the distinct developmental potential for A-/G+ and A+/G+ cells."

2. *In Figure 1 legend, it describes panel D but there is no panel D there.*

→ We apologize to the reviewer and have corrected this mistake.

3. *In the flow cytometry studies, information should be provided as to the survival of cells through the isolation and labeling procedure. Were dead cells excluded? This is important because there is always the possibility that certain cell populations are more hardy and more likely to survive the*

isolation and labeling procedures and thus bias the results one way.

→ We apologize that the use of Propidium Iodide (to exclude dead cells) has not been obvious. We have now modified the text accordingly.

From the manuscript: "For the gating, we used side-scatter vs. forward-scatter to determine neural cells. Cell debris and dead cells were identified by Propidium Iodide (Miltenyi Biotec) and excluded from the analyses."

4. *In the grafting studies, while we are told that a high % of cells survived the grafting process, no information is provided what this actually means. Was this 90% or 9%? This should be corrected.*

→ We agree with the reviewer that that sentence was not appropriate and may create confusion. Therefore, we decided to delete it from the text as it is technically impossible to provide an estimation of survival cells after grafting due to a known high inter-animal variability. We have analyzed a total of 2721 cells in 21 different cerebella with a median number of 75 cells per cerebellum in a single section series. However, we could not find in the literature a reference value to which compare the number of engrafted cells and evaluate the quality of our grafting. Therefore, we prefer to avoid any attempt of grafting quality evaluation.

5. *At the same time, it was not immediately clear what age the donor mice were receiving the grafts. This should be spelt out to make it clear.*

→ We have now explicitly indicated in the text the age of donor mice, as follows "A+/G+ or A-/G+ cells were injected into the cerebellar vermis of P3 β -actin-GFP- littermates"

6. *Please correct page 12, line 313, change "extend" to "extent."*

7. *Same page, line 318, change "on" to "at."*

→ We thank the reviewer for these corrections and have modified them in the manuscript.

2nd Editorial Decision

Decision Letter

Dear Dr Jungblut:

Thank you for submitting your manuscript to the Journal of Neuroscience Research. We have now received the reviewer feedback and have appended those reviews below. I am glad to say that the reviewers are overall very enthusiastic and supportive of the study. The statistics editor did raise some minor concerns and made some suggestions for clarification, but I expect that these points should be relatively straightforward to address. If there are any questions or points that are problematic, please feel free to contact me. I am glad to discuss.

We ask that you return your manuscript within 15 days. Please explain in your cover letter how you have changed the present version and submit a point-by-point response to the editors' and reviewers' comments. The journal has adopted the "Expects Data" data sharing policy, which states that all original articles and reviews must include a Data Availability Statement (DAS). Please see <https://authorservices.wiley.com/author-resources/Journal-Authors/open-access/data-sharing-citation/data-sharing-policy.html#standardtemplates> for examples of an appropriate DAS. Please include the DAS in the manuscript as well.

If you require longer than 15 days to make the revisions, please contact Dr Junie Warrington (jpwarrington@umc.edu). To submit your revised manuscript: Log in by clicking on the link below <https://wiley.atyponrex.com/submissionBoard/1/cd0c4602-9c34-4608-92f8-20c3c3b8f1e6/current>

If the above link space is blank, it is because you submitted your original manuscript through our old submission site. Therefore, to return your revision, please go to our new submission site here [submission.wiley.com/jnr](https://www.submission.wiley.com/jnr) and submit your revision as a new manuscript; answer yes to the question "Are you returning a revision for a manuscript originally submitted to our former submission site (ScholarOne Manuscripts)? If you indicate yes, please enter your original manuscript's Manuscript ID number in the space below" and including your original submission's Manuscript ID number (jnr-2020-Dec-9286) where indicated. This will help us to link your revision to your original submission.)

Thank you again for your submission to the Journal of Neuroscience Research; we look forward to reading your revised manuscript.

Best Wishes,

Dr Stephen Crocker
Associate Editor, Journal of Neuroscience Research

Dr Junie Warrington
Editor-in-Chief, Journal of Neuroscience Research

Editorial Comments to the Editor:

- Figure S1 is still very small and has numerous panels. Consider breaking these up into 3 separate figures (a-c).

Associate Editor: Crocker, Stephen

Comments to the Author:

Reviews were consistent their positive appraisal of this resubmitted study. However, specific issues with data presentation and analysis were raised that should be addressed with additional explanations and modifications to data presentation in a revised manuscript.

Statistics Editor: McArthur, David

Comments to the Author:

Sorry but the tiny sample sizes in Figures 3b and c have the potential to be highly problematic unless you have strong evidence beforehand that $n=3$ is sufficient to address that question. Limitations section must address this in detail. The JNR Questionnaire Item 7 cannot legitimately be rendered as 'N/A'.

Another reviewer has asked about the choice of first vs last five CSs. What statistical and/or research design considerations caused you to elect to drop half of the (presumably) diligently collected datapoints to leave only this particular subset? Be clear in your logic, as the choice should not be arbitrary, nor should it be "folks always omit data". (Note that the statistics could well differ depending on the full dataset.)

Reviewer: 1

Comments to the Author

The authors have thoroughly answered all required revisions and edits, and I have no further issues with the manuscript. It was a pleasure to review this manuscript and work with the authors to improve upon the already great quality of the work.

Reviewer: 2

Comments to the Author

All points raised in my original review have been addressed satisfactorily with the exception of the typo on the new page 14 line 329 which remains. "on" should be switched for "at."

Authors' Response

Decision on Manuscript # jnr-2020-Dec-9286

We would like to thank the editors and reviewers for the overall positive feedback and the valuable comments.

As follows we will address the comments of the editor and reviewers point by point.

Comments from editors and reviewers

Editorial Comments to the Editor:

- Figure S1 is still very small and has numerous panels. Consider breaking these up into 3 separate figures (a-c).

We have now split Figure S1 in 3 new figures (Figure S1-S2-S3) and modify the corresponding figure legends and text accordingly.

Associate Editor: Crocker, Stephen

Comments to the Author:

Reviews were consistent their positive appraisal of this resubmitted study. However, specific issues with data presentation and analysis were raised that should be addressed with additional explanations and modifications to data presentation in a revised manuscript.

We have addressed the additional issues as follows:

Statistics Editor: McArthur, David

Comments to the Author:

Sorry but the tiny sample sizes in Figures 3b and c have the potential to be highly problematic unless you have strong evidence beforehand that $n=3$ is sufficient to address that question. Limitations section must address this in detail. The JNR Questionnaire Item 7 cannot legitimately be rendered as 'N/A'.

We appreciate and understand the criticism raised by the Statistics Editor. However, in our experience “n=3 mice” is an appropriate sample size used in the field of brain development. In our case, this sample size was sufficient to reach a statistically significant difference between the two populations of interest. In particular, for the experiments relative to figure 3b,c we have analyzed a significant amount of cells (at least 10'000 cells) for sample, as it is

routinely done for these type of flow cytometry analyses. Phenotypical and functional differences between ACSA+ and ACSA2- cell populations were further confirmed by additional experiments and multiple approaches in the rest of the study.

In addition, all our experiments involved animals and therefore we had to comply with the Principle of the 3R for animal experimentation (Replace, Reduce, Refine) that recommends to reduce the number of laboratory animals to the greatest possible extent and use only as many animals as are needed to obtain a statistically significant outcome.

The JNR Questionnaire Item 7 was already answered as follows: “Page 3, Paragraph 2; Page 23, Paragraph 3-4 (Outlook)”. The answer N/A refers to Item 6 but the document layout was misleading. We apologize for the misunderstanding. We have now modified the questionnaire and made it more clearly legible.

Another reviewer has asked about the choice of first vs last five CSs. What statistical and/or research design considerations caused you to elect to drop half of the (presumably) diligently collected datapoints to leave only this particular subset? Be clear in your logic, as the choice should not be arbitrary, nor should it be "folks always omit data". (Note that the statistics could well differ depending on the full dataset.)

We have analyzed all the samples collected without omitting any data point. The differences in the sample size are due to different numbers of animals available and therefore collected in different experiments. If the reviewer is referring to Figure 3, please note that each graph is a different flow cytometry experiment in which we have analyzed all the collected samples.

Reviewer: 1

Comments to the Author

The authors have thoroughly answered all required revisions and edits, and I have no further issues with the manuscript. It was a pleasure to review this manuscript and work with the authors to improve upon the already great quality of the work.

We also would like thank the reviewer for his/her help in improving our manuscript with very valuable comments.

Reviewer: 2

Comments to the Author

All points raised in my original review have been addressed satisfactorily with the exception of the typo on the new page 14 line 329 which remains. "on" should be switched for "at."

We thank the reviewer for the positive feedback and for spotting the typo. We have now changed the “on” into “at”.

3rd Editorial Decision

Decision Letter

Dear Dr Jungblut:

Thank you for submitting your manuscript "ACSA-2 and GLAST classify subpopulations of multipotent and glial restricted cerebellar precursors" by Jungblut, Melanie; Kantzer, Christina; Parmigiani, Elena; Cerrato, Valentina; Tomiuk, Stefan; Knauel, Michail; Buffo, Annalisa; Bosio, Andreas.

You will be pleased to know that your manuscript has been accepted for publication. Thank you for submitting this excellent work to our journal.

In the coming weeks, the Production Department will contact you regarding a copyright transfer agreement and they will then send an electronic proof file of your article to you for your review and approval.

Please note that your article cannot be published until the publisher has received the appropriate signed license agreement. Within the next few days, the corresponding author will receive an email from Wiley's Author Services asking them to log in. There, they will be presented with the appropriate license for completion. Additional information can be found at <https://authorservices.wiley.com/author-resources/Journal-Authors/licensing-open-access/index.html>

Would you be interested in publishing your proven experimental method as a detailed step-by-step protocol? Current Protocols in Neuroscience welcomes proposals from prospective authors to disseminate their experimental methodology in the rapidly evolving field of neuroscience. Please submit your proposal here: <https://currentprotocols.onlinelibrary.wiley.com/hub/submitaproposal>

Congratulations on your results, and thank you for choosing the Journal of Neuroscience Research for publishing your work. I hope you will consider us for the publication of your future manuscripts.

Sincerely,

Dr Stephen Crocker
Associate Editor, Journal of Neuroscience Research

Dr Junie Warrington
Editor-in-Chief, Journal of Neuroscience Research

Associate Editor: Crocker, Stephen
Comments to the Author:
(There are no comments.)

Authors' Response

4th editorial decision

Decision Letter

Author response